# Aqueous alternating electrolysis prolongs electrode lifespans under harsh operation conditions

Jie Liang[1,2], Jun Li[2], Hongliang Dong[3], Zixiaozi Li[2], Xun He[2], Yan Wang[2], Yongchao Yao[2], Yuchun Ren[2], Shengjun Sun[1], Yongsong Luo[1], Dongdong Zheng[1], Jiong Li[4], Qian Liu[5], Fengming Luo[6], Tongwei Wu[2]✉, Guang Chen[7]✉, Xuping Sun[1,2,6]✉ & Bo Tang[1,8]✉

It is vital to explore effective ways for prolonging electrode lifespans under harsh electrolysis conditions, such as high current densities, acid environment, and impure water source. Here we report alternating electrolysis approaches that realize promptly and regularly repair/maintenance and concurrent bubble evolution. Electrode lifespans are improved by co-action of Fe group elemental ions and alkali metal cations, especially a unique $Co^{2+}$-$Na^+$ combo. A commercial Ni foam sustains ampere-level current densities alternatingly during continuous electrolysis for 93.8 h in an acidic solution, whereas such a Ni foam is completely dissolved in ~2 h for conventional electrolysis conditions. The work not only explores an alternating electrolysis-based system, alkali metal cation-based catalytic systems, and alkali metal cation-based electrodeposition techniques, and beyond, but demonstrates the possibility of prolonged electrolysis by repeated deposition-dissolution processes. With enough adjustable experimental variables, the upper improvement limit in the electrode lifespan would be high.

How to create and optimize a portfolio of solutions to eventually decarbonize the current energy landscape, which depends heavily on fossil fuels, is one of the primary challenges of our time[1-3]. Electricity generated from renewable energy sources can be used to drive electrochemistry processes, which would play a key role in transforming fossil fuel-based technologies into more sustainable ones in the future[4-6]. Noticeably, the electrosynthesis of $H_2$ from water is sufficiently sustainable and desirable compared to traditional natural gas reforming process[7,8], but many significant challenges remain for the current water electrolysis technologies, particularly the limited

lifespan of the electrodes under the harsh electrolysis conditions (HECs). Such HECs frequently occur in real-world water electrolysis applications[9-17], such as the scouring of the electrodes by aggressive bubbles releases at high ampere-level current densities ($J$), electrode dissolution due to localized electrolyte acidity, and impurity interference from impure water sources (i.e., halides ions). For instance, proton exchange membrane water electrolyzers (PEMWEs) that use pure water as the feeding electrolyte suffer from local acidity, which leads to the dissolution of the anode[18-20]. Notwithstanding that many metals are highly fragile in acidic media[21], Ir still represents the most

[1]College of Chemistry, Chemical Engineering and Materials Science, Shandong Normal University, Jinan, Shandong, China. [2]Institute of Fundamental and Frontier Sciences, University of Electronic Science and Technology of China, Chengdu, Sichuan, China. [3]Center for High Pressure Science and Technology Advanced Research, Shanghai, China. [4]Shanghai Synchrotron Radiation Facility, Shanghai Advanced Research Institute, Chinese Academy of Sciences, Shanghai, China. [5]Institute for Advanced Study, Chengdu University, Chengdu, Sichuan, China. [6]Center for High Altitude Medicine, West China Hospital, Sichuan University, Chengdu, Sichuan, China. [7]Shaanxi Key Laboratory of Chemical Additives for Industry, College of Chemistry and Chemical Engineering, Shaanxi University of Science & Technology, Xi'an, Shaanxi, China. [8]Laoshan Laboratory, Qingdao, Shandong, China. ✉e-mail: twwu77@uestc.edu.cn; chenandguang@163.com; xpsun@uestc.edu.cn; tangb@sdnu.edu.cn

stable metal for acidic oxygen evolution reaction (OER). The last few years have witnessed a rapid development of various Ir-based materials including $IrO_x/SrIrO_3$ film[22], $La_2LiIrO_6$[23], pyrochlore oxides $R_2Ir_2O_7$ (R = Ho, Tb, Gd, Nd, and Pr)[24], $IrNi@IrO_x$[25], $Ta_{0.1}Tm_{0.1}Ir_{0.8}O_{2-\delta}$[26], $IrO_x \cdot nH_2O$[27], etc., for OER at low pH conditions. However, the genuine catalyst option for anode in PEMWEs, highly pricey Ir, is going to limit a global rollout of terawatt-scale PEM devices[28]. Moreover, previous work confirmed apparent performance loss following a reduction in the Ir loading of MEA[29]. Even though many studies have focused on developing Ru-based electrodes as the alternative[30–32], the expensive cost of noble metals like Ru cannot afford their widespread use.

As such, a great deal of studies have focused on prolonging the lifetime of non-noble metal-based electrodes under HECs. In fact, despite remarkable advances[33–37], a suitable noble-meal-free material that does not dissolve under acidic oxidation conditions does not appear to exist. In this regard, Simonov and co-workers reported a prolonged electrode lifespan in acidic OER via continuous deposition–dissolution of solid oxides on a fluorine-doped tin oxide substrate or a Pt/Ti substrate[38]. To be specific, catalyst dissolution occurs at all times during electrolysis in the low-pH electrolyte, whereas catalysts are formed in situ from dissolved $Co^{2+}$, $Fe^{3+}$, and $Pb^{2+}$, i.e., the dissolution can be offset more or less by the deposition (Fig. 1a). The method is quite effective; however, the electrolysis time at industrial-level $J$ is limited (< 8 h). Moreover, Pb does improve the stability of the coating, but it is also a highly toxic pollutant. Although many past reports that aim at improving the electrode lifetime under HECs have certain limitations, the findings have continued to refine and enrich our knowledge about building a longer-term, non-noble metal-based electrolysis system. Therefore, more potential strategies for prolonging the electrode lifespan under HECs are highly desirable.

Here, we showcase a strategy to enhance the lifetime of the electrode under HECs only using non-noble metals, which is realized by periodically, promptly, and evenly constructing a protective outer layer for the electrode. To illustrate the effectiveness of the strategy, we used a commercial Ni foam electrode (NiFE), which is not resistant to acid or halogen ions, as the substrate. An in situ protect concept (Fig. 1b) can even make the NiFE electrolyze in low pH solution solidly for 93.8 h at the high alternating $J$ of −2 and 2 A cm$^{-2}$ (superior to many $O_2$ evolution electrocatalysts in acids; see details in Supplementary Table 1). Note that the lifespan of NiFE achieved in our system is ~ 47 times that of NiFE in the conventional water electrolysis system, and AE alone cannot accomplish this dramatic boost in lifespan. The critical point is that a special co-action of $Co^{2+}$ and $Na^+$ adaptively realizes fast and uniform protective coating growth on the Ni core. Theoretical

calculations under working conditions further reveal in-depth mechanisms of how $Co^{2+}$ cooperates with alkali metal cations (AMCs), including $Li^+$, $Na^+$, and $K^+$, especially $Na^+$, to help the Ni electrode survive longer under HECs. (I) Synergistic effects of hydrated $Na^+$ with higher diffusion ability, (II) moderate interactions between $Na^+$ and metal Co, and (III) suitable adsorption strengths of metal Co are responsible for more prolonged Ni electrode survival. In addition, since ample $Na^+$ naturally exists in natural water sources (like seawater), our preliminary findings in this work are even more promising to be further developed.

## Results

### Conventional water splitting issues & why alternating electrolysis

Generally, conventional low-temperature water electrolysis (CLTWE) systems show several limitations. (1) Experimental environments for CLTWE systems are ultrapure water-based electrolytes (i.e., highly caustic 30% KOH solution for alkaline water electrolyzers, pure water for PEMWEs, and 1 M KOH solution for immature anion exchange membrane water electrolyzers)[39], however, such systems are not directly applicable to impure water sources. In addition, even if generating $H_2$ from water permits a small water demand (~ 9 kg $H_2O$/kg $H_2$)[3], the net zero emissions scenario in the future will result in an enormous rise in the volume of freshwater consumed for water splitting-based activities. As such, innumerable freshwater-stressed regions may rely on direct electrolysis of low-grade natural briny water, while solids, microbes, and other impurities (e.g., chlorides in seawater) from local water sources would more or less impair the achievable performances of current CLTWE technologies, leading to challenging issues like more rapid catalyst corrosion (e.g., triggered by chlorine chemistry) and membrane instability. (2) Currently, the membrane electrode assembly (MEA) design in PEMWEs is expensive and burdensome to manufacture, and the durability and associated maintenance issues increase the capital cost of electrolyzers. Studies in techno-economics predicted that capital costs of PEMWEs with high device complexity would account for nearly all expenses associated with electrolytically splitting water to $H_2/O_2$ when electricity prices decline (even be free) in a renewable energy future[40,41]. It appears that making breakthroughs in CLTWE technologies is challenging, particularly under HECs. Therefore, solely targeting performance optimization of existing CLTWE technologies (acting as plan A) would not be enough. For water splitting-based electrochemistry processes in the future, (i) innovative electrolysis strategies[42], (ii) the exploration of unconventional electrolysis systems (UESs), e.g., methodological advances in water electrolysis methods or designs of

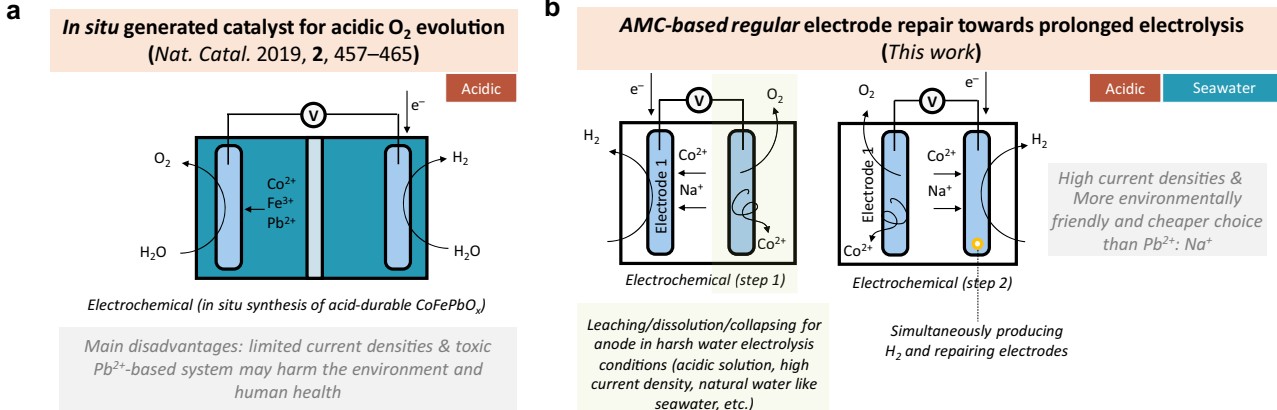

**a** *In situ* generated catalyst for acidic $O_2$ evolution (*Nat. Catal.* 2019, **2**, 457–465)

Electrochemical (in situ synthesis of acid-durable $CoFePbO_x$)

*Main disadvantages: limited current densities & toxic $Pb^{2+}$-based system may harm the environment and human health*

**b** *AMC-based regular* electrode repair towards prolonged electrolysis (*This work*)

Electrochemical (step 1)

Electrochemical (step 2)

*High current densities & More environmentally friendly and cheaper choice than $Pb^{2+}$: $Na^+$*

*Leaching/dissolution/collapsing for anode in harsh water electrolysis conditions (acidic solution, high current density, natural water like seawater, etc.)*

*Simultaneously producing $H_2$ and repairing electrodes*

**Fig. 1 | Schemes showing two different strategies toward prolonging the electrode lifespan under HECs. a** One representative work by Simonov and the co-workers. Adding $Co^{2+}$, $Fe^{3+}$, and $Pb^{2+}$ into the electrolyte to achieve simultaneous acidic water electrolysis and deposition of dissolved ions. **b** An AMC-based electrolysis scheme for artificially repairing electrodes to perform prolonged operation under HECs (e.g., acidic environments, high $J$, natural water sources).

disruptive electrolyzers with lower costs[38,43–51], and/or (iii) fundamental advances would be crucial and provide more possible alternative and complementary countermeasures (acting as back-up plan X). Various reported UESs in recent years, like the work by Simonov's group (Fig. 1a)[38], can be found in Supplementary Fig. 1 and Supplementary Table 2.

Note that this work is not meant to replace or compete with existing technologies like PEMWEs, which have their own application scenarios. Instead, this work may act as a critical reference/model for later studies that develop membraneless electrolysis systems, valuable alternating current (a.c.)-based electrolysis processes, AMC-based electrocatalytic systems, and AMC-based electrodeposition techniques for preparing advanced materials, etc. To date, little research has been done on pulsed a.c.-based water electrolysis due to the generation of explosive $H_2/O_2$ mixtures as well as water self-heating at high frequencies. A few cases in organic electrosynthesis applying the a.c.-based electrolysis method have been documented, which showed better results than direct current (d.c.)-based electrolysis[52,53]. Moreover, even in aqueous solution, pulsed electrochemistry-mediated systems have been widely reported, such as pulsed $CO_2$-to-multi-carbon products process[54], pulsed nitrate-to-ammonia electroreduction[55], C – N coupling reaction to arylamines[56], AE-based regeneration of alkaline trapping liquids for direct air capture[57], and AE-based uranium extraction from seawater[58]. Supplementary Note 1 contains more pulsed electrochemistry studies in recent years. In addition, the shift in direction is what distinguishes d.c. from a.c.; hence, a.c. can be considered as d.c. for a sufficiently short period of time. When polarity changes of the a.c.-based electrolysis are slow enough (i.e., at a rather low a.c. frequency), the a.c.-based electrolysis can be treated almost as d.c.-based electrolysis. Therefore, the probable $H_2/O_2$ mixing and water self-heating occurring at high polarity change frequencies can be avoided by lowering the frequency.

## In situ protect concept based on $Fe^{2+}$, $Co^{2+}$, $Ni^{2+}$, and $Mn^{2+}$

Considering unavoidable metal leaching/dissolution in HECs (acid etching, strongly oxidative environments, violent gas release, etc., see Fig. 2a), we believe that we can just let the outer layer of the electrode dissolve freely, but we will fix/repair it after a period of electrolysis to prolong the service life of the electrode. One of the effective electrolytic modes that can achieve such an operation is a.c.-based electrolysis. We propose and comprehensively verify a repairing approach of in situ regular electrode maintenance (Figs. 1b, 2b, c, Supplementary Fig. 2, and Supplementary Note 2) to try to achieve a longer electrolysis time. When $O_2$ is evolved at a metallic electrode (i.e., during the OER process), the metallic ion species will leach out and dissolve, while metal ions in the electrolyte will be deposited on the electrode during the hydrogen evolution reaction (HER). In this study, prolonging water splitting in an uncommon way is demonstrated in an acidic solution (Fig. 2) using a three-electrode system. Ions of the VIII group in the periodic table (i.e., $Fe^{2+}$, $Co^{2+}$, $Ni^{2+}$) were chosen for demonstration in that such Fe group elements are all active for water electrolysis. In fact, the cyclic electrolysis can theoretically continue for an unusually long time (maybe even permanently) once the equilibrium is established between deposition and dissolution of coating, but in practice, it is difficult to achieve. We provide detailed reasons for the final degradation of performance as well as why permanent alternating electrolysis cannot be achieved at this stage (Supplementary Note 3). Due to the technological limitations, the cycle of deposition-dissolution processes is not completely reversible, but we demonstrate the great potential of this strategy. NiFE that easily dissolves is preferably chosen as the substrate to find the best metal ions for effective coating formation in acids. It also has a 3D fully penetrating network structure and is well-known for its good intrinsic OER activity. The coating/armor that forms in situ will protect the NiFE from fast dissolution. When the electrolyte is too acidic, the repair is almost ineffective in that there is

no rapid deposition of metal ions (Supplementary Fig. 3). Moreover, changing the step time length or the applied potential for repair steps (Supplementary Fig. 4), as well as adding extra $Ni^{2+}$ into the electrolyte (Supplementary Figs. 5, 6), cannot improve the electrode lifetime at low solution pH. This difficulty in achieving deposition in low pH environments can be found in a previous excellent study as well[38]. Since it is hard for ions to be quickly deposited under extremely low pH, we carried out AE experiments after increasing the pH higher than 1. Note that after a minor pH increase (∼ 0.3 to ∼1.5), the surface of the NiFE was repaired successfully (Supplementary Fig. 7). Figure 2d–f shows voltage-time ($V$–t) curves for various ion combinations under the ampere-level $J$ (1 A cm$^{-2}$ & − 1 A cm$^{-2}$). Lifetime improvements achieved by increasing the concentration of a certain ion are rather limited (panel I in Fig. 2d), yet the combination of different ions is more effective in enhancing the coating lifetime (panels III and IV in Fig. 2d), except that the combination of $Mn^{2+}$ and $Ni^{2+}$ leads to a negative effect (panel II in Fig. 2d). The $Ni^{2+}$-$Fe^{3+}$ combo leads to significant applied potential oscillations at the later stage of electrolysis (panel III in Fig. 2d) because NiFE dissolves but keep forming dendrites (Supplementary Fig. 8). Due to the good potential of Mn for stabilizing OER-active materials in acidity and its relatively low price[59], the effects of $Mn^{2+}$ on the AE performance of Fe group elemental ions are investigated. Interestingly, $Mn^{2+}$ is identified to enhance the coating lifespan in acid containing two kinds of cations of Fe group elements (Fig. 2e), and its concentration can be optimized to sustain electrolysis for 14.2 h (panel IV in Fig. 2e). The duration far exceeds the anodic electrolysis time of NiFE in acids without metal ions (14.2 h versus 2.1 h, panel I in Fig. 2d), indicating the effectiveness of the repairing strategy. The effect of $Mn^{2+}$ on coating lifespan in $H_2SO_4$ is quite different in $H_3PO_4$ (pH is ∼ 1.17 prior to the test). The cycling curves suggest that $Mn^{2+}$ fails to boost the coating lifespan at all, and adding $Mn^{2+}$ to the electrolyte containing $Ni^{2+}$ and $Co^{2+}$ reduces its durability (panel IV in Fig. 2f). As shown in panel III in Fig. 2f, the activity of NFE in solution with 0.1 M $Ni^{2+}$ and 0.1 M $Co^{2+}$ is inferior to that in solution with 0.1 M $Ni^{2+}$ and 0.1 M $Fe^{3+}$, whereas the electrode lifespan is much better in the solution with 0.1 M $Ni^{2+}$ and 0.1 M $Co^{2+}$. Noticeably, in situ generated coating in electrolytes containing only $Ni^{2+}$ and $Co^{2+}$ is able to support 14 h of electrolysis in acidic environments. Such high durability in materials for acid water splitting is rare since the majority of state-of-the-art noble-metal-free catalysts can be rapidly deactivated at the $J$ of 1 A cm$^{-2}$. Obviously, a more sustained lifespan for NiFE can be realized with a.c. electrolysis in a solution containing specific ions (Fig. 2g).

## Co-Na interaction to prolong electrode lifespan

Our preliminary investigations ascertained that AE in electrolytes with certain metal ions can even boost the operating life of the highly vulnerable electrode, metallic Ni foam, under high $J$ and low pH conditions. Through more efforts, we again boosted the electrolysis time and operating $J$ of NiFE to higher levels under harsh electrolytic conditions simultaneously. As shown in Fig. 3a, the combinations of AMCs and Fe group element ions in acidic media under ultrahigh $J$ of 2 and − 2 A cm$^{-2}$ are evaluated and compared systematically. Unlike ions like $Mn^{2+}$, AMCs including $Li^+$, $Na^+$, and $K^+$ exhibit a special role in the protection of NiFE under acidic water electrolysis conditions by profoundly facilitating the Fe group element ions-based in situ electrode maintenance. The introduction of AMCs changes the evolution process of coating on the Ni substrate. In $H_2SO_4$ media containing $Fe^{3+}$ and $Li^+$, bubbles evolve so violently that although NiFE can get repaired, the attached armor falls off quickly (photo series 1 in Fig. 3b). A similar breakdown of the protective armor is observed for the combo of $Co^{2+}$ and $Li^+$ (photo series 2 in Fig. 3b). Although the two coatings are formed and collapsed differently, neither is strong enough. The $Ni^{2+}$-$Li^+$ combo provides weaker protection to the Ni substrate than the $Fe^{3+}$-$Li^+$- and $Co^{2+}$-$Li^+$-based counterparts, and the corresponding Ni core dissolves fast and completely (photo 3 in

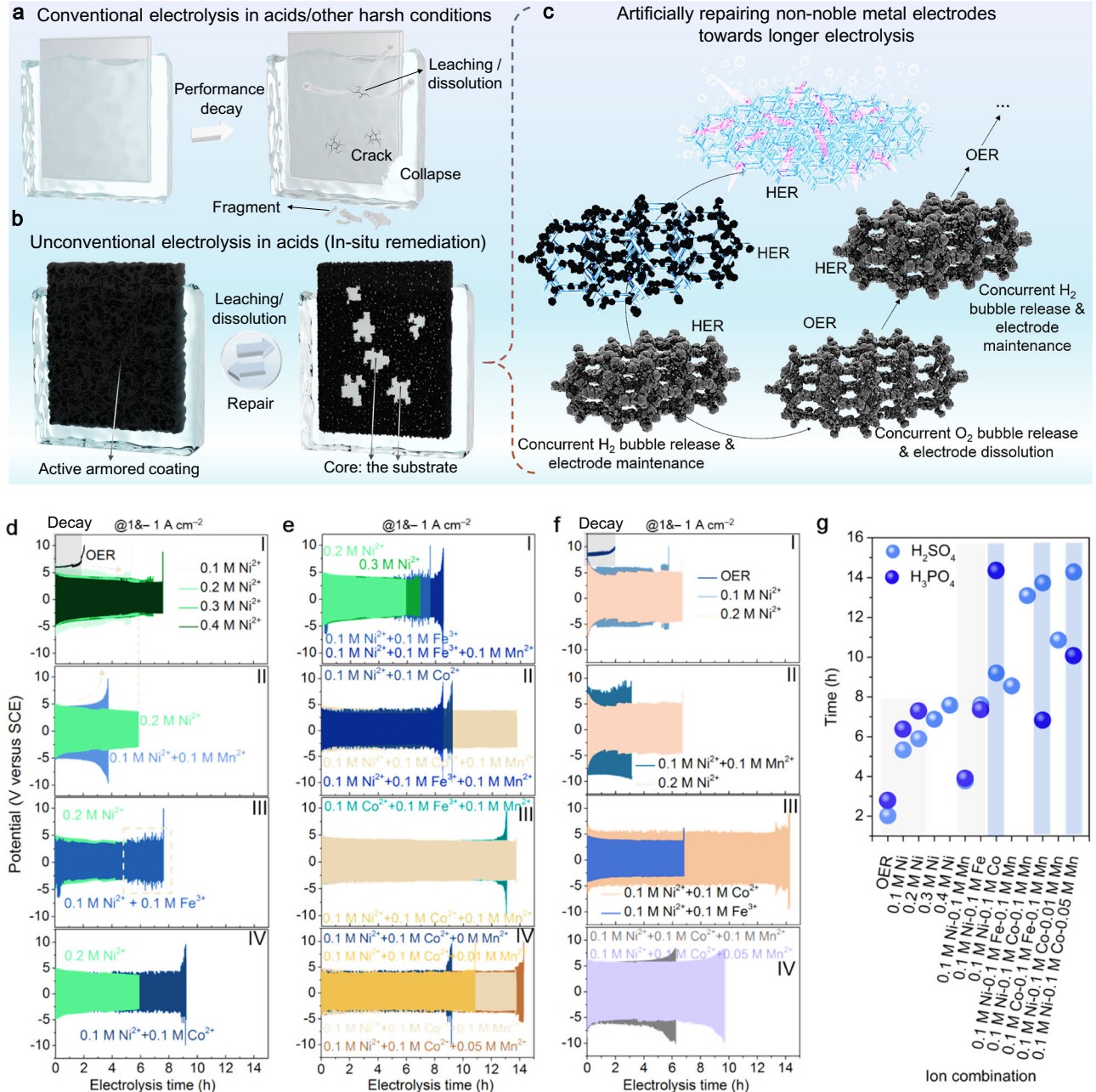

**Fig. 2 | Preliminary electrode lifetime optimization studies in acid solution using various non-noble metal ions at $J$ of −1&1 A cm⁻².** **a** Almost unavoidable metal leaching/dissolution in conventional electrolysis systems at HECs versus (**b**) timely electrode repair in cyclic electrolysis. The latter may, in theory, support everlasting electrolysis operation. **c** A further sketch explanation of the electrolysis mode. The electrode will synthesize $H_2$ while adsorbing metal ions from the solution for on-site repair. **d** Electrolysis in case of increasing $c_{Ni2+}$ and using a combination of two different ions (pH > 1 $H_2SO_4$). **e** Electrolysis based on combinations of three different ions (pH > 1 M $H_2SO_4$). **f** Electrolysis under various conditions (pH > 1 $H_3PO_4$). **g** Adjusting the concentration of ions and combining suitable ions to prolong acidic water electrolysis. Metallic Ni foam that is acid non-resistant is adopted as an example.

Fig. 3b). This implies that if the coating is not built in time, the electrodes will soon collapse. Even though using $Fe^{3+}$, $Co^{2+}$, and $Li^+$ at the same time can further improve the electrolysis time, the coating that is not robust enough will eventually come off the NiFE surface (photo series 4 in Fig. 3b). The protective armor deposited during the in situ repair periods falls apart due to massive bubble releases under high $J$ OER operation, thus indicating that a better adhesion of the armor to the substrate is highly desirable. Moreover, the classical Fe-Ni synergistic effect for water splitting is also reflected in our test results. For instance, the combo of $Fe^3$, $Ni^{2+}$, and $Li^+$ achieves much longer electrolytic time and lower overpotentials than those of the $Fe^{3+}$-$Li^+$ combo and $Ni^{2+}$-$Li^+$ combo (Fig. 3a). This combination ($Fe^3$,

$Ni^{2+}$, and $Li^+$) is in favor of generating dendrites around the NiFE, but still cannot completely cover the inner Ni core (photo series 5 in Fig. 3b). While simultaneously using $Co^{2+}$, $Ni^{2+}$, and $Li^+$ also increase the lifespan of the NiFE, the outside coating is likewise unstable and continuously degrading during the electrolysis process (photo 6 in Fig. 3b). In fact, AE processes maybe roughly divided into three stages for many ion combinations, namely the deposition area (fast growth of the protective coating) at the beginning, subsequent equilibrium area (coating growth almost equal to leaching/dissolution) and final dissolution area (coating growth slower than leaching) that may occur. Such dissolution may occur in any of the following ways: coating dissolution, substrate dissolution, or both. In solutions

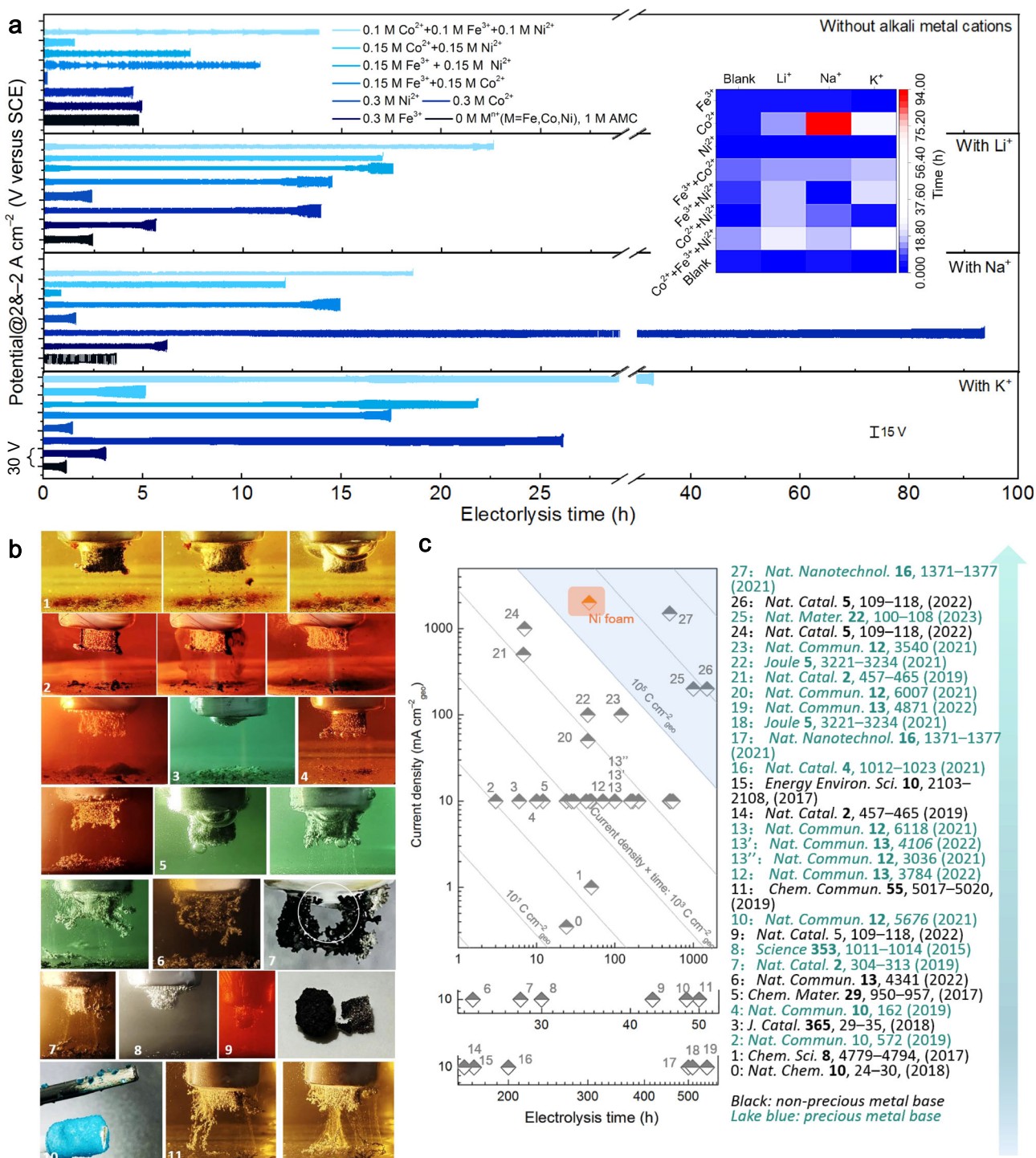

**Fig. 3 | Fe group element ions assisted by AMCs in sustaining the acidic water electrolysis more powerfully at the *J* of − 2&2 A cm⁻².** **a** Alternating electrolysis based on Fe group elemental ions and the electrolysis with the addition of different AMCs. The inset image is a heat map visualizing the unique combination of Na⁺ and Co²⁺. **b** Optical photos captured under various reaction conditions. The photos are not micrographs, so there are no corresponding scale bars. **c** Comparison of electrode lifespan in low pH solution (< 3).

containing $Fe^{3+}$ $Co^{2+}$, $Ni^{2+}$, and $Li^+$, the coating is prone to forming from the NFE edge, which exposes the center of the NFE core to an acidic environment and causes slow corrosion. Eventually, only the outside coating is left, which cannot continue electrolysis since the coating is almost disconnected from an inner conducting plate of the electrode holder (image NO.7 in Fig. 3b). Interestingly, simply adding AMCs to the electrolyte only leads to faster electrode dissolution. In an $H_2SO_4$ solution containing only $Li^+$, the electrode quickly releases

bubbles, but it quickly corrodes and becomes smaller (image NO.8 in Fig. 3b). For the best ion combo of $Co^{2+}$ and $Na^+$, even though electrolysis ended, the electrode is not actually dissolved; rather, only the joint place where the electrode exposed to the electrolyte is connected to the part inside the electrode holder is broken (image NO.9 in Fig. 3b), thereby indicating that a capillary action makes part of the acidic electrolyte to seep into the joint place and cause the self-fracture. The ion combo of $Ni^{2+}$ and $K^+$ leads to the precipitation

of a large number of blue crystals (image NO.10 in Fig. 3b), and the electrolytic time is not long. In the solution with equal amounts of $Fe^{3+}$, $Co^{2+}$, and $Ni^{2+}$ but no AMCs, observable dendrites (image NO.11 in Fig. 3b) are formed between 1600 s and 3000 s during the reaction. Such dendrites are not firm. In short, high-stable operation in acids via the in situ regular electrode maintenance requires the coating to meet the following characteristics: (i) be built quickly, (ii) nice adhesion to the substrate, (iii) uniform and full coverage of the core substrate, and (iv) defensive and protective ability of the coating being adequate for maintaining dissolution-deposition balance. In the presence of AMCs, the NiFE is able to electrolyze more consistently at high $J$, which is far longer than NiFE tested with no AMCs and NiFE tested without repairing steps. A heatmap based on data obtained from all the optimization combinations (inset of Fig. 3a) reflects enhancement trends more visually. AMCs significantly improve the electrolysis time, and NiFE lifetime enhancement based on the $Na^+$-$Co^{2+}$ pair is even more pronounced. Under low pH conditions, the lifespan of the acid-soluble Ni electrode achieved in this work (Fig. 3c) exceeds that of the many advanced OER catalysts available today. This is a breakthrough. In the presence of $Na^+$, we investigated the effects of various ratios of Fe group elemental ions on the electrolysis performance (Supplementary Fig. 9). Different initial $Co^{2+}$/$Fe^{3+}$ ratios, including 0:1, 1:2, 1:1, 2:1, and 3:1, achieve roughly the same levels of electrolysis time (all below 15 h), which are significantly shorter than that of the 1:0 ratio without adding $Fe^{3+}$ (93.8 h), indicating that the introduction of $Fe^{3+}$ is less able to affect the performance of $Na^+$-$Co^{2+}$ pair. Similarly, $Ni^{2+}$ has no ability to positively affect the $Na^+$-$Co^{2+}$ pair as the electrolysis time increases gradually with increasing $Co^{2+}$/$Ni^{2+}$ ratios. The synergy between $Ni^{2+}$ and $Fe^{3+}$ exists, with the most stable combination being when the $Ni^{2+}$ to $Fe^{3+}$ ratio is 2:1. In addition, we provide the data of Fig. 3a in the uncompressed form to show more details. Supplementary Figs. 10–13 present the potentials measured versus SCE, and Supplementary Figs. 14–17 present the potentials against reversible hydrogen electrode (RHE). In order to verify the reproducibility of the critical result of 93.8 h, we measured the lifespan of NiFE at the optimal $Co^{2+}$-$Na^+$ combo corresponding to 93.8 h several times and provided completely independent experimental results (Supplementary Fig. 18). The average electrolysis time is still maintained at over 90 h. Furthermore, when the concentration of $Co^{2+}$ was reduced to 0.2 M, and the $J$ changed to −1&1 A $cm^{-2}$, the final electrolytic lifespan of NiFE can still exceed 90 h (Supplementary Fig. 19). These experiments illustrate that under the conditions we have identified, it is possible to extend the electrolysis lifespan of NiFE to more than 90 h by the synergistic action of $Co^{2+}$ and $Na^+$. Importantly, we reduced the concentrations of $Co^{2+}$ (60 mM) and $Na^+$ (0, 200, 400, 800 mM) and investigated the stability of different $Na^+$-$Co^{2+}$ ratios. The promoting effect of $Na^+$ on the stability of $Co^{2+}$-based a.c. electrolysis is still present at low concentrations (Supplementary Fig. 20a). In addition, we varied the step time for the a.c. electrolysis of the $Na^+$-$Co^{2+}$ combo, and the longer step time results in a significantly increased electrolysis time (Supplementary Fig. 20b), again illustrating the potential for further exploitation of this AE. The interaction between $Co^{2+}$ and $Na^+$ during a.c. electrolysis leads to in situ formation of a black coating on the NiFE, which is displayed in Supplementary Fig. 21. The coating under different testing conditions (i.e., after 10, 20, 30, and 50 h of AE tests under ampere-level $J$) was investigated in detail using scanning electron microscopy (SEM) (Fig. 4a), ex-situ X-ray diffraction (XRD) (Fig. 4b), depth-dependent X-ray photoelectron spectroscopy (XPS) (0, 15, and 35 nm in depth) (Fig. 4c–e), transmission electron microscopy (TEM), and high-resolution TEM (HRTEM) (Supplementary Figs. 22–24), etc.,. We provided all XPS survey spectra, C 1$s$ spectra, and the original data in Supplementary Figs. 25–27. Moreover, we indirectly studied the interaction of three AMCs with $Co^{2+}$ under different conditions by

electrochemical voltammetry characterizations (Supplementary Figs. 28–35). In addition, we recorded the electrochemical impedance spectroscopy (EIS) data. Compared to Nyquist plots of NiFE before electrolysis (Supplementary Fig. 36a), the EIS semicircle becomes smaller after a 10-min OER process (Supplementary Fig. 36b), indicating the anodic bias can oxidize the NiFE, but this semicircle change is not as significant as the change after the HER process. Well-separated semicircles can be observed for the NiFE after a 10-min HER process under different testing parameters (Supplementary Fig. 36c). This suggests that there is a marked difference between the changes made to the electrodes by HER and OER electrolysis alone. Besides, running at the anodic potentials does not introduce cobalt-containing catalytic materials. In addition, the data in Supplementary Fig. 36d–h clearly demonstrates that longer AE testing times would lead to smaller EIS semicircles, confirming faster charge transfer processes. After a sufficiently long period of tests, when the AC signal passes through the electrode, diffusion control will override the electrochemical control in the low-frequency portion, and the Warburg impedance will appear (Supplementary Fig. 36h). The fitting data for sample recorded after 10-min HER testing, after 10-min AE testing, samples after 1.25-h AE testing, sample after 3-h AE testing, and the sample after 22.8-h AE testing can be found in Supplementary Fig. 37. The conductivity is not a decisive factor in determining the efficacy of the electrode lifespan enhancement. To demonstrate this, we measured and roughly compared solution resistance values of three different sets of electrolytes (Supplementary Fig. 38), namely the electrolyte containing 1 M $Na^+$ and 0.3 M $Co^{2+}$, the electrolyte containing 1 M $K^+$ and 0.3 M $Co^{2+}$, and the electrolyte containing 1 M $Na^+$ and 0.3 M $Ni^{2+}$. Based on these ion combinations, the resulting electrode lifespan varies considerably. Noticeably, the solution resistance values corresponding to the three ion combinations do not differ significantly from each other (Supplementary Fig. 38c). In addition, we performed double-layer capacitance ($C_{dl}$) measurements to study the changes in the electrode. The $C_{dl}$ value of the electrode does rise after a short period of electrolysis (Supplementary Fig. 39), indicating an increase in the electrochemical active surface area (ECSA). This actually matches the results of the SEM characterization and electrochemical data, where the protective coatings with catalytic activity appear on the substrate surface after electrolysis based on the $Co^{2+}$-$Na^+$ combo. The enlarged ECSA should lead to better electrochemical activity. X-ray absorption fine structure (XAFS) data of three samples, including NiFE after 5 min of HER testing (i.e., deposition step only) under $J$ of −2 A $cm^{-2}$ (denoted as HER only), NiFE after 30 h of AE operation under $J$ of −2 A $cm^{-2}$ and 2 A $cm^2$ (denoted as After 30 h), and NiFE after 50 h of AE operation under $J$ of −2 A $cm^{-2}$ and 2 A $cm^2$ (denoted as After 50 h), are shown in Fig. 4f–g. Co K-edge X-ray absorption near edge structure (XANES) profiles show gradually increased adsorption edges following the trend of "After 30 h" > "After 50 h" > "HER only" > $Co_3O_4$ (Fig. 4f). The sample "HER only" displays the most reduced Co valence state amongst the catalysts (+2.37), while the other catalysts obtained after 30 h or after 50 h of operation present a Co valence state of +2.9 and +3.143 (Fig. 4g). The valence state of +2.37 indicates that the surface of the "HER only" electrode is extremely susceptible to oxidation even when tested only under the HER step. Fourier-transformed (FT) spectra of the extended X-ray absorption fine structure (EXAFS) curves of both "After 30 h" and "After 50 h" samples do not match well the reference $Co_3O_4$ and CoO. The near +3 valence state of Co for both "After 30 h" and "After 50 h" samples suggest the formation of CoOOH. In short, due to the continuous deposition-oxidation (i.e., dissolution)-deposition-oxidation (dissolution) processes during the AE tests, the resulting coating may consist of crystalline Co, Co (oxy)hydroxides, amorphous $Na_xCo_yO_z$, etc., accommodating evenly distributed $Na^+$ dopants (from mapping results in Fig. 4a), and the coating is

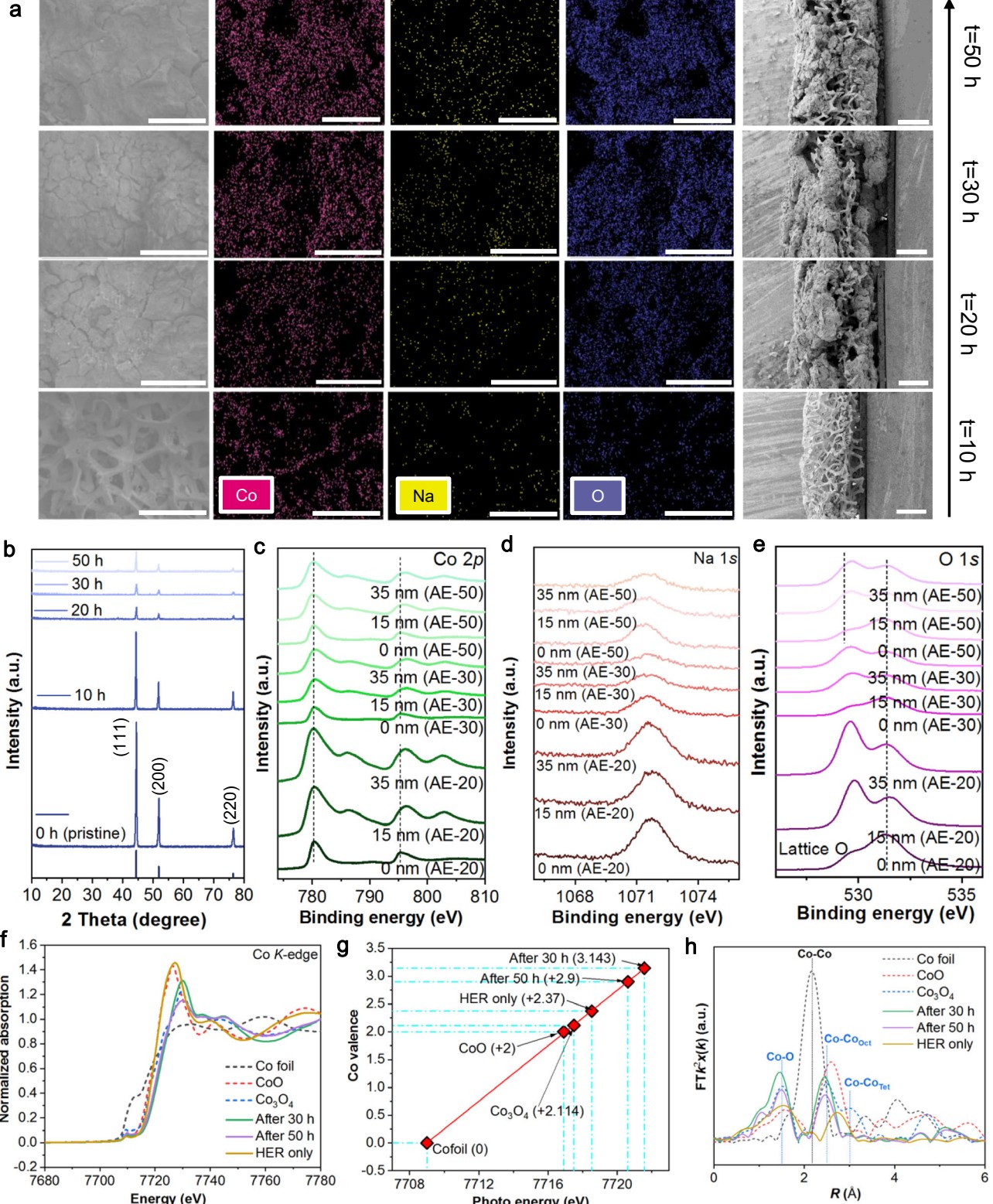

**Fig. 4 | Characterization of the surface coating on the electrode.**
**a** Morphologies, element distribution, and cross-sectional views. Scale bars for the SEM images and the corresponding elemental mapping images are 250 μm, and scale bars for the cross-sectional SEM images are 600 μm. **b** XRD patterns of the different electrodes reflecting the ultra-low crystallinity nature of the coating. XPS analysis in (**c**) Co 2*p* region, (**d**) Na 1*s* region, and (**e**) O 1*s* region. **f** Co K-edge XANES profiles. **g** Summary of Co oxidation states for all the investigated samples. Co foil, CoO, and Co₃O₄ are standards for comparison. **h** Fourier transform (FT) of the Co K-edge EXAFS spectra.

catalytically active (from the cyclic voltammetry results in Supplementary Figs. 28–33, 35) and thermodynamically stable. Moreover, we also demonstrate the potential feasibility of the AMC-based AE strategy for the PEM-based device (Supplementary Fig. 40). In addition, the synergistic interaction $Na^+$ and $Co^{2+}$ that we have discovered under the pulsed electrolysis conditions may have informative and positive impacts in other electrocatalytic fields (Supplementary Note 1). For instance, enhanced OER stability in conventional acidic water oxidation catalysis might be realized by preparing catalytic materials containing both Co and Na. We also verified this from experiments (Supplementary Fig. 41).

## More insights into the process and relevant mechanisms

It is experimentally confirmed that the introduction of $Na^+$ into the electrolyte optimally allows a uniform Co-containing protective layer to form on the electrode surface in a shorter period of time. Therefore, it is of critical importance to understand the reason why $Na^+$ controls the formation of a protective layer in a fast and uniform manner. Based on these unknowns, we perform state-of-the-art constant-potential molecular dynamics (MD) simulations (CPMD) to explore them using grand canonical ensemble density functional theory (GCE-DFT), which can more realistically simulate the atomistic phenomena occurring at electrocatalyst-electrolyte interfaces under working conditions[60–64]. Specifically, the GCE-DFT method allows proper treatment of the electrode potential while properly accounting for solvent interactions; molecular dynamics (MD) simulation is a powerful tool to study the dynamic evolution of the structural and electronic properties[62]. Their couplings have made it available to observe features of real electrocatalyst/electrolyte interfaces, as detailed in Supplementary Figs. 42, 43.

Results show that when the electrode potential is set to – 3 V versus standard electrode potential ($V_{SHE}$) in the dynamic CPMD simulation (Fig. 5a), the $H_2O$ molecules are not adsorbed on Co clusters. It suggests that Co metal exists in a low valence state at the negative potential. Then, we added $Na^+$ into the interface and found that $Na^+$ forms a hydrated state with three water molecules (Fig. 5b)[65]. Currently, the metal Co also remains in the lower valence states due to the reduction potential applied. Next, we apply a positive potential of $U = 3 V_{SHE}$, and observe that the $Na^+$ ions are hydrated with four water molecules, while two water molecules are adsorbed on the metal Co, see Fig. 5c. It suggests that metal Co is easily oxidized by oxygen atoms in water at applied positive potential, thus forming the higher oxidation states. Note that the adsorbed water pulls the Co atoms away from the protective layer due to the longer Co-Co distance with 2.05 Å (Fig. 5c), suggesting that the dissolution of Co at positive potential in the experiment is primarily caused by the traction effect caused by the adsorption of $H_2O$. Furthermore, we also calculate the interface environments of $K^+$ and $Li^+$. Results show that when $U = – 3 V_{SHE}$, $K^+$ forms a hydrated state with three water molecules (Fig. 5d), and $Li^+$ forms a hydrated state with one water molecule (Fig. 5e). Based on the results, we calculate the root-mean-square displacement (RMSD) to investigate why the protective layer of Co is formed more evenly after adding AMCs ($Li^+$ $Na^+$ and $K^+$) in the experiment. It can be seen that the RMSD importantly increases after adding AMCs (Fig. 5f), indicating that the diffusion capacity of interface molecules or atoms is improved. In comparison, we can infer that the increase in diffusion capacity is due to the formation of a hydrated state of AMCs. Jiang Group has recently also confirmed the information that the hydrated $Na^+$ has a higher diffusion capacity[66], so we believe that the increased diffusion rate due to the formation of hydrated AMCs can improve the deposition of metal Co at the interface. It should be noted, however, that there is no significant difference between $Na^+$, $Li^+$, and $K^+$ for RMSD, which cannot explain another important finding in the experiment that the uniform of the Co-containing protective layer is optimally improved after the introduction of $Na^+$ into the electrolyte. As a

result, we further examined the interaction between AMCs and Fe group element ions. Results show that $Li^+$ and Fe group element ions have an interaction energy that approaches zero, indicating a weak interaction, see Fig. 5g. It is also evident from the corresponding differential charge density diagram that the charge density of the $Li^+$ is low (Fig. 5h), thus there is no obvious electrostatic interaction between it and the Fe group element ions. Then, we observe that the interaction energy is increased between $Na^+$ and Fe group element ions (Fig. 5g), which can be attributed to the increase in electron density of $Na^+$, thereby increasing the electrostatic interaction between it and Fe group element ions. The differential charge density for $K^+$ indicates that the electron density around $K^+$ continues to increase, resulting in a stronger electrostatic interaction between it and Fe group element ions and, consequently, a higher interaction energy see Fig. 5g, h. We also examine the adsorption energies ($E_a$) of Fe, Co, and Ni atoms on Ni (111), and results show that the Co atom exhibits higher $E_a$, see Supplementary Fig. 44. It suggests that the Co atom is preferable to those of Fe and Ni atoms to deposit on Ni (111), and thus forms a protective layer more easily, consistent with experiment results. Moreover, different shapes of Co clusters are also considered, and the results show that the flat Co cluster prefers to form than that three-dimensional Co cluster after adding $Na^+$, suggesting the uniform growth of Co protective coating on the Ni (111), as detailed in Supplementary Fig. 45.

Accordingly, we speculate that both (i) the moderate interaction energy of $Na^+$ with Co and (ii) the available adsorption strength of Co on Ni (111) are the reasons why it can maximize the uniform deposition of Co. Overall, the calculations under working conditions clearly reveal that the high deposition and dissolution rates of Co after adding $Na^+$ under alternating current are due to the diffusion ability of hydrated AMCs, the moderate interaction between $Na^+$ and Co and the suitable adsorption strength of Co on the electrode material. Finally, the mechanisms are summarized in Fig. 5i.

## Prolonged electrolysis in seawater

Direct splitting impure brine water with a simple apparatus is of high significance for the global deployment of electrochemical $H_2O$-to-$H_2$ conversion technologies. On the basis of the impressively enhanced lifespan of fragile metallic Ni substrate under ampere-level $J$ and low pH water electrolysis conditions, natural seawater (without filtration/purification) is further adopted as the electrolyte for confirming again the great effectiveness of in situ regular electrode maintenance strategy. The electrolysis time of NiFE in seawater does not decrease much after adding $H_2SO_4$ (Supplementary Fig. 46); both electrodes are completely dissolved in less than 100 s, indicating that etching of NiFE by chlorine (electro)chemistry and impurities in seawater is considerably greater than that by acidic solution. Even in the presence of the $Na^+$-$Co^{2+}$ pair, NiFE still dissolved completely after ~ 1100 s, and the time was increased by more than 5 times compared to electrolysis without adding $Co^{2+}$. This might be due to the fact that (i) Ni dissolves rapidly at high $J$ in seawater (the main reason) and (ii) the impurity interference greatly slows down the pace of protective coating growth. Accordingly, we adopted a titanium mesh electrode (TiME) that was affordable and durable as the substrate for AE in seawater. Noticeably, TiME is unable to maintain the AE on its own in (acidic) seawater as it lacks the sufficient activity needed to oxidize anions like $Cl^-$ (Supplementary Fig. 47). Moreover, electrolysis of acidic seawater does not result in precipitation. In a solution with the $Na^+$-$Co^{2+}$ pair that affords the longest electrolysis lifespan, using TiME substrate allows for more sustained electrolysis (Supplementary Fig. 48), but the coating generates rather slowly on the TiME. Unacidified seawater is thus used for electrolysis; however, coating on the TiME grows too fast and becomes too thick to gradually lower the performance with increasingly higher electrode potentials (inset of Supplementary Fig. 49). Moreover, adding $Fe^{3+}$ or $Ni^{2+}$ leads to poor lifespan in seawater, neither exceeding 600 s. Even at a relatively low $J$ of 0.2 A cm$^{-2}$, the potentials required

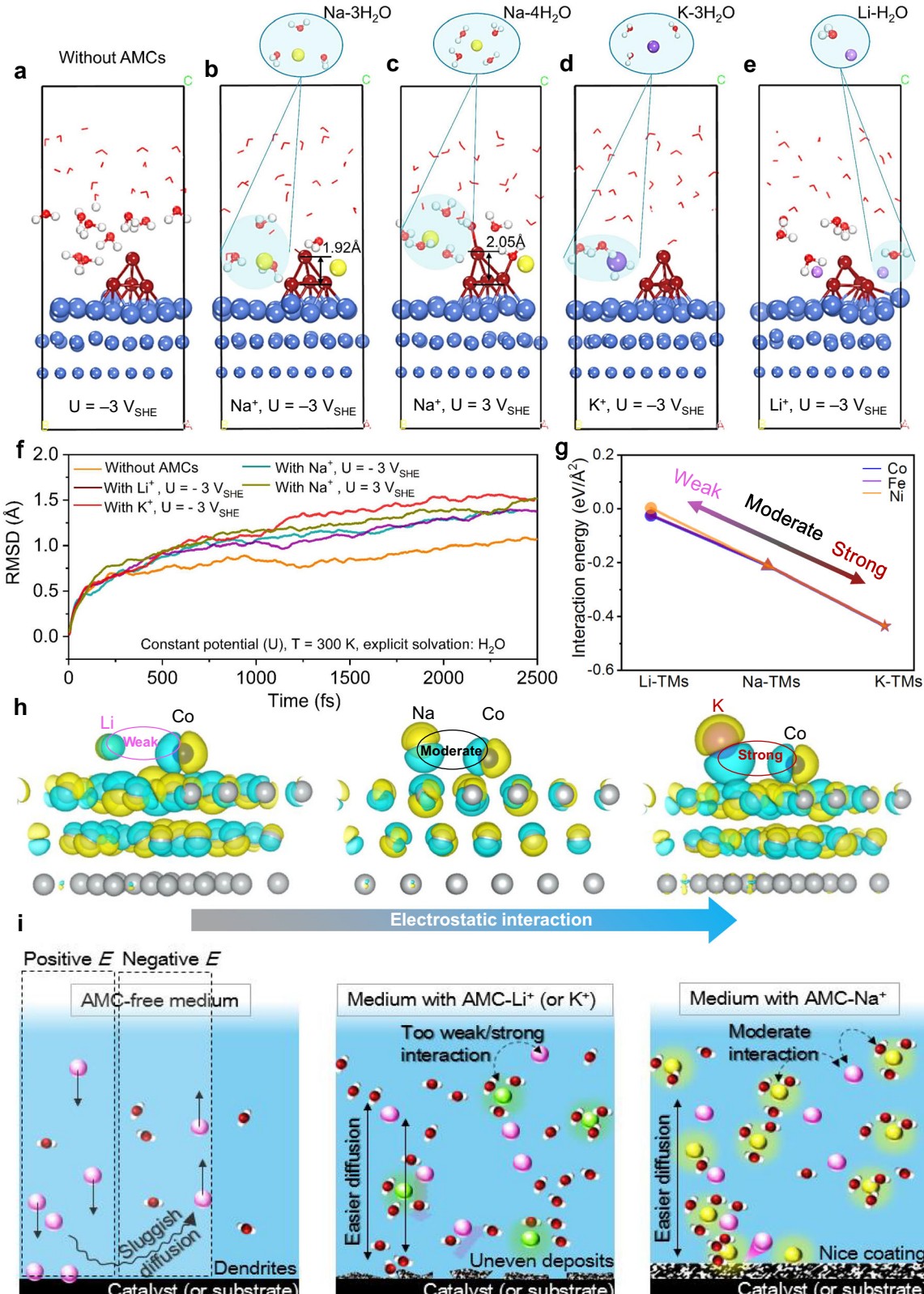

**Fig. 5 | In-depth understanding of surface reaction processes. a–e** Interfaces of the catalyst and electrolyte with/without AMCs. Co: brown; Ni: blue; O: red; H: white; Na⁺: yellow; K⁺: purple; Li⁺: light purple. **f** RMSD with/without AMCs at a constant potential. **g** Interaction energies between the AMCs and the TMs (Fe group element ions in this work). **h** Differential charge density between AMCs and Co. Blue and yellow topologies denote electron depletion and accumulation, with an isosurface value of 0.02 e/Å³. **i** Simplified diagrams.

are quite high (Supplementary Fig. 49). Interestingly, when seawater contains the same molarity of $Ni^{2+}$ and $Fe^{3+}$, the electrolytic potential suddenly decreases sharply. The lifespan increases rapidly (Supplementary Fig. 50a), and the surface-derived dendrite structure is exceptionally stable. Only the TiME is responsible for $O_2$ during the OER period, while both the TiME and dendrites are producing $H_2$ during the HER (fig. Supplementary Fig. 50b). This dendrite has HER activity, and dissolves slowly in acidic seawater. Under OER catalysis, part of the electrode does not bubble, so the required potential is large; under HER catalysis, two parts bubble together, so the potential is smaller. Notably, due to the change in the geometric area of the electrode in the later period of electrolysis, the $J$ will also change accordingly. We explored the influence of the ratio of $Ni^{2+}/Fe^{3+}$ (1:2 and 2:1) on electrolysis, and the best performance is achieved at the ratio of 1:1 (Supplementary Fig. 51). Note that the dendritic structures can be dissolved in acid and recycled (Supplementary Fig. 50c). Despite the good durability achieved by the $Ni^{2+}$-$Fe^{3+}$ combo (Supplementary Fig. 50a, the electrolysis is deliberately stopped, and the Fe-group element (Supplementary Fig. 50b) deposits form predominantly at the edges of the time. We further alter the acidity of $Co^{2+}$-containing seawater (Supplementary Fig. 50d) in an effort to control the development of the surface coating. In moderately acidic seawater, electrolysis continues despite large $J$ fluctuations. We point out again that the electrode leaches out (i.e., deposition < leaching) under an overly acidic condition, but surplus growth of deposits under near-neutral conditions (or too much metal ions) may cause $H_2$ FE decay (Supplementary Fig. 52). In Supplementary Fig. 50e, f the generated deposit structure shows little visual change after the stabilization of the sediment (mostly the shiny and gray metallic Co), indicating that electrolysis starts to enter a relatively stable state and the coating on the electrode may not collapse or accumulate, i.e., deposition (repairing) almost equals dissolution. In addition, some metallic Co deposits that are not oxidized but not adhered strongly enough will fall off and accumulate, and such deposits can also be recovered (Supplementary Fig. 50f), which further indicates the sustainable potential and value of our strategy. At the same time, the surfaces of the time and the graphite rod both produce the black protective coating containing Co and Na elements (Supplementary Fig. 50g).

## AE-based water electrolysis

Compared to d.c.-based electrolysis, which has been the mainstream electrolytic scheme for over two centuries, a.c.-based electrolysis is underexplored, and a great deal of recent research has shown that we may use a.c.-based electrolysis schemes to solve some of the existing problems encountered in the field of d.c.-based electrolysis as well as to develop new systems[52–58]. With a.c.-based electrolysis, we can have more options than just a static redox environment (static means constant potential or current), allowing for a wide variety of electrolytic environments with more adjustable parameters. At the same time, however, a.c.-based water electrolysis means that the same electrode surface would alternately produce hydrogen and oxygen. In this regard, we present potential solutions (Fig. 6) for future work. Inside the electrolyzer, an autonomous gas separation unit is incorporated (see basic working principle in Fig. 6a), which will be embedded in the water. At the same time, we are designing a device for AE that automatically changes the collection outlet (Fig. 6a, b), which is a further step toward ensuring safe production. Moreover, this risk of gas mixing at the same electrode will diminish as the frequency of polarity shifts decreases as the technology improves.

## Discussion

In summary, this work aims to find and preliminarily investigate a strategy for long-term water splitting under HECs encountered in real-world electrolysis systems. In situ, regular/routine maintenance based on pairings of AMCs and Fe group ions, especially the $Na^+$ and $Co^{2+}$,

promptly and evenly creates structures functioning as armored plating to improve electrode lifespan, a significant roadblock in acidic OER. With the special co-action of $Co^{2+}$-$Na^+$, a fragile commercial Ni foam that easily dissolves in acids is able to sustain electrolysis with aggressive bubble release in acidic solutions for nearly 100 h (93.8 h). Theoretical calculations under working conditions reveal that the hydrated $Na^+$ with higher diffusion ability facilitates the metal Co diffusion, while both the moderate interaction energy of $Na^+$ with Co and the available adsorption strength of Co on Ni (111) regulate the uniform deposition of Co. Their synergistic effects are responsible for longer Ni electrode survival under HECs. Our results demonstrate the great potential of AE based on effective ion combinations to be developed as an unconventional technique for practical HECs with extended electrode lifespan, especially in the context of decreasing electricity costs following the future widespread deployment of renewable energy sources. Our work not only develops aqueous AE-based systems, AMC-based electrocatalytic systems, and AMC-based electrodeposition techniques and beyond but also demonstrates the possibility of long-time electrolysis under HECs by effective, repeated deposition-dissolution processes. With enough adjustable variables of AE-based systems[52–58], the upper limit of the improvements in the electrode lifespan or even the product selectivity would be high.

## Methods

### A list of chemicals/materials

In this work, chemicals were used directly without any purification process. We point out that due to the long experimental period of this work (~2 years), some of the reagents purchased were from different manufacturers and batches. For some chemicals, like $K_2SO_4$, we used products from different manufacturers, but the results were not significantly different. Sulfuric acid ($H_2SO_4$, 95.0% ~ 98.0%) was purchased from a local chemical manufacturer, Chengdu Chron Chemical Co., Ltd. Phosphoric acid ($H_3PO_4$, ≥ 85.0%) was also purchased from a local chemical manufacturer, Chengdu Jinshan Chemical Test Co., Ltd. Perchloric acid ($HClO_4$, 70.0% ~ 72.0%) was purchased from Chengdu Jinshan Chemical Test Co., Ltd. Lithium sulfate monohydrate ($Li_2SO_4 \cdot H_2O$, AR, 99.0% ~ 100.5%) was purchased from Tianjin Kermel Chemical Reagent Co., Ltd. Sodium sulfate ($Na_2SO_4$, AR, 99%) was purchased from Shanghai Aladdin Biochemical Technology Co., Ltd. Potassium sulfate ($K_2SO_4$) was purchased from Tianjin Fuchen Chemical Reagent Co., Ltd (AR, ≥ 99.0%) as well as Chengdu Jinshan Chemical Test Co., Ltd (AR, ≥ 99.0%). Iron(III) sulfate ($Fe_2(SO_4)_3$) was obtained from Shanghai Yuanye Bio-Technology Co., Ltd (AR, 21% ~ 23% for Fe) and Chengdu Chron Chemical Co., Ltd (AR). Manganese sulfate ($MnSO_4$, AR, 31% ~ 35% for Mn) was purchased from Shanghai Macklin Biochemical Co., Ltd. Cobalt sulfate heptahydrate ($CoSO_4 \cdot 7H_2O$) was obtained from Chengdu Chron Chemical Co., Ltd (AR, ≥ 99.5%) as well as Shanghai Macklin Biochemical Co., Ltd (AR, ≥ 99.0%). Nickel sulfate hexahydrate ($NiSO_4 \cdot 6H_2O$) was obtained from Chengdu Chron Chemical Co., LTD (AR, ≥ 98.5%) as well as Tianjin Fuchen Chemical Reagent Co., Ltd (AR, ≥ 98.5%). Potassium chloride (KCl) was purchased from Shanghai Macklin Biochemical Co., Ltd. We used a water purifier (UPT-l-10T) from Sichuan ULUPURE Ultrapure Technology Co. Ltd. (website: https://www.ccdup.com/) for the purification of water, and ultrapure water ($18.25 \, M\Omega \, cm^{-1}$) was used throughout the experiments. Photos of the chemical reagents used in this work are also provided in Supplementary Fig. 53. Nickel foam (Supplementary Fig. 54a–c) with a thickness of 2.0 mm was purchased from Kunshan LONGSHENGBAO Electronic Materials Co., LTD (the company website: http://www.ksslsb.com/). Titanium mesh with a purity of > 99.5%. (Supplementary Fig. 54d, e) was purchased from Taobao.com. In addition, all of our reference electrodes used in this work are commercial reference electrodes (Supplementary Fig. 55). In this work, saturated calomel electrodes (SCE, CHI 150) as the consumables are stored in a 50-mL centrifuge tube containing saturated

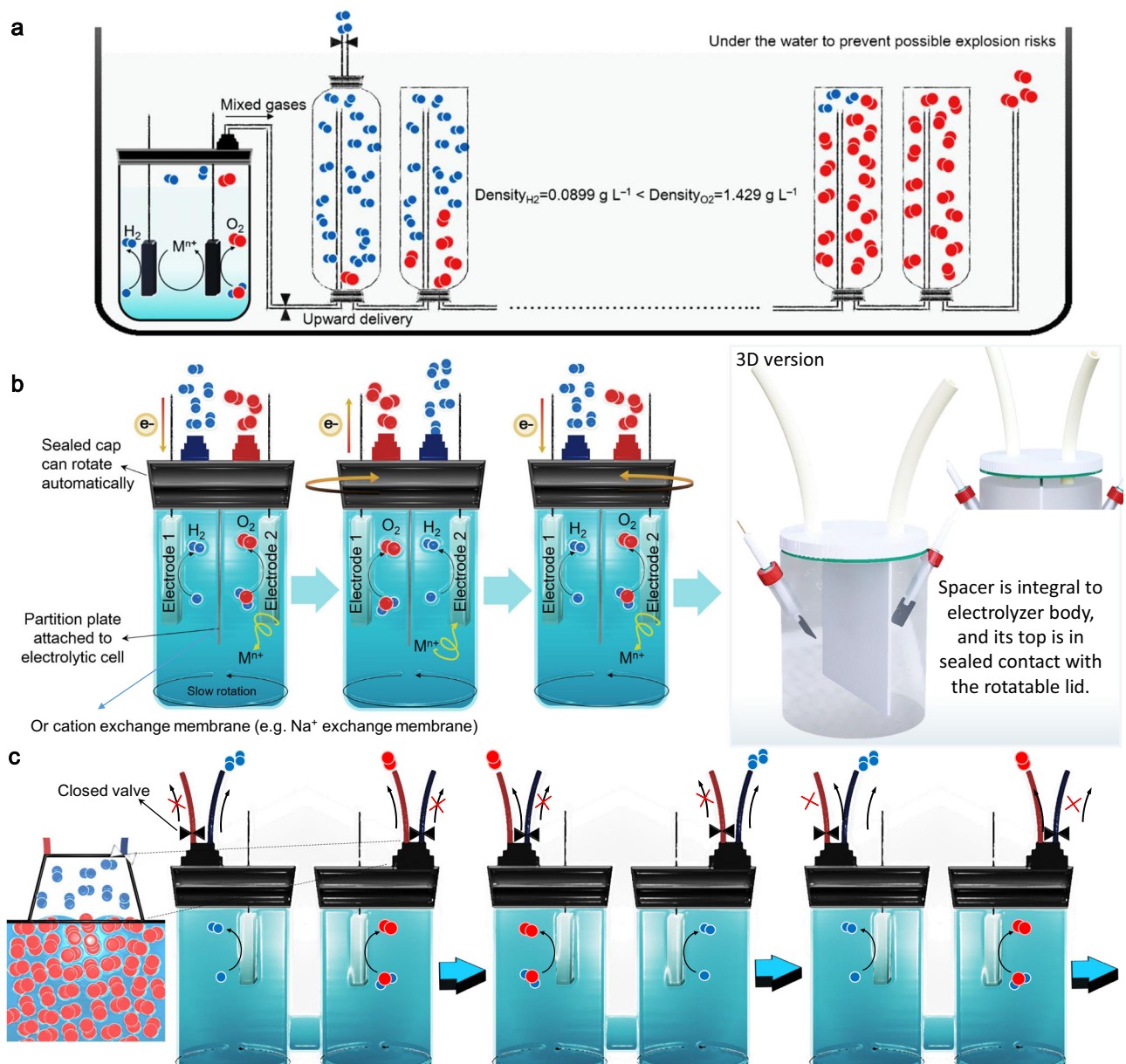

**Fig. 6 | Possible design principles for gas separation in the a.c. electrolysis-based device in the future work. a** Schematic illustration of a possible gas separation way based on cyclic electrolysis in future works (Adoption of simple/understandable devices as examples). Continuous separation of $H_2$ and $O_2$ in an air-free environment (e.g., underwater, using plastic or glass devices) to obtain a higher concentration of $H_2$ (must be over the flammability limit). **b** Rotating the cap reserves a specific outlet for the outflow of the corresponding type of gas. The single-chamber cell is separated by a spacer in the middle, but not completely separated into two chambers (or divided by membrane). **c** Two valves control which outlet the created gas during a specific period exits through. These operations to separate the gas are done underwater and require coupling additional separation units.

potassium chloride solution when not in use for a short period of time (e.g., in two or three days), while we would cover and seal the bottom of the electrode (i.e., the ceramic liquid junction) with a plastic cap when not in use for a longer period of time. Not only do we need to maintain the moisture content of the ceramic core, but we also need to prevent crystallization from clogging the ceramic core.

## Material characterizations

Electrolytes will remain on the electrode surface after an electro-chemical test. Thus, the electrode after the test was rinsed briefly and slowly with ultrapure water and dried in an oven (50 °C) as a sample for further characterization. All the XRD data were collected using a measurement method of stepping with a start angle of 10 degrees and a termination angle of 80 degrees. The step angle was 0.02 degrees.

The tube voltage was 40 kV, and the tube current was 30 mA. The data were collected at room temperature (~ 20 °C). We cut a piece of the electrode into smaller sizes (e.g., ~ 0.2 cm × ~ 0.2 cm) and pasted them onto conductive tapes on the sample stage for normal SEM characterization. For capturing the cross-section views, the electrode was cut along the middle and then fixed vertically to the sample stage with the cross section facing upwards after being gently purged clean with argon gas. A magnetron sputtering apparatus was used to spray platinum on the surface of the sample (operation time: ~ 3 min) to improve the image quality before proceeding with characterizations. The vacuum of the electron microscope sample chamber is generally below $5 \times 10^{-5}$ mbar before the collection of images. The accelerating voltage for SEM testing and energy dispersive spectrometer testing was 5 kV and 10 kV, respectively. A piece of dried electrode sample was

cut into smaller pieces, which were immersed in ethanol solution for prolonged sonication. Then, the electrode pieces were taken out to obtain colloid suspension for preparation of TEM characterization samples (dripping nearly 10 μL solution on Cu grids/carbon). In the first step, a low vacuum level is realized by a mechanical pump, which reduces the pressure from atmospheric pressure to a pressure of 1.33 to 0.133 Pa (0.01 to 0.001 mmHg). Next, a higher vacuum level was realized by an oil diffusion pump, and the pressure was reduced from the low vacuum level to a pressure ranging from $13.33 \times 10^{-4}$ to $10^{-5}$ Pa ($10^{-5}$ to $10^{-6}$ mmHg). In addition, the transmission electron microscope was operated at an accelerating voltage of 200 kV. The microscope was operated at an accelerating voltage of 120 kV. Samples for the XPS characterization (Thermo ESCALAB 250XI adopting Mg as the excitation source) are kept in nitrogen-filled centrifuge tubes sealed with tape before the testing. Data acquisition at different depths is realized based on the operation of different argon sputtering times. The etch depth is converted relative to the specimen and is not the true depth of the sample. The analysis pressure for XPS was $2.42 \times 10^{-9}$ mBar, and the preparation pressure was $3.04 \times 10^{-8}$ mBar. XAFS spectra at the Co K-edge were recorded at the BL11B beamline of the Shanghai Synchrotron Radiation Facility (SSRF). The beam current of the storage ring was 200 mA in a top-up mode. The incident photons were monochromatized by a Si(111) double-crystal monochromator, with an energy resolution $\Delta E/E \sim 1.4 \times 10^{-4}$. The spot size at the sample was $\sim 200 \, \mu m \times 250 \, \mu m$ (H × V). The energy calibration was performed using a Co reference foil. The reference spectra were measured in the transmission mode, with the ionization chambers filled with $N_2$.

## Electrochemical measurements

Acidic electrolytes, including diluted $H_2SO_4$, $H_3PO_4$, and $HClO_4$ solutions. The solution will be replaced with a newly made one if it is not used up within three days. Moreover, different metal sulfates including $Li_2SO_4 \cdot H_2O$, $Na_2SO_4$, $K_2SO_4$, $MnSO_4$, $Fe_2(SO_4)_3$, $CoSO_4 \cdot 7H_2O$ and $NiSO_4 \cdot H_2O$ are quantitatively added to the acidic solution for AE. In short, the electrolyte is $H_2SO_4$ solution (or $H_3PO_4$ solution), and different metal sulfates can be added to it as reactants. We provide a photo-based flowchart (Supplementary Fig. 56) to enable the readers to better understand the screening experiments. Unless otherwise noted, about 60 mL of acidic electrolyte was used as the electrolyte. We adopt inexpensive and readily available substrates like commercial NiFE and TiME for experiment demonstration for more groups to replicate our findings and to conduct more varied research based on this work. The temperature of the electrochemical experiment was conducted at around 25 °C. The scan rate used for the cyclic voltammetry tests was 20 mV s$^{-1}$ with a sensitivity of 0.1 A V$^{-1}$. Impedance measurements were carried out in different electrolytes without stirring. The high frequency was fixed at 100,000 Hz, and the low frequency was fixed at 0.01 Hz. The initial E was equal to the open-circuit potential for each test, and the amplitude was fixed at a relatively low value of 0.003 V. Electrochemical experiments were measured using a CHI 660E workstation (or CHI 760E) and a three-electrode system. We employed a commercial graphite rod (replaced on a regular basis) as the counter electrodes to exclude unwanted interference with metal leaching from the metal counter electrodes. The reference electrode, saturated calomel electrode (SCE) or Ag/AgCl, is also replaced after a certain period of electrolysis. No *iR* correction is performed because the open circuit potential actually changes all the time, and the electrochemical test is kept at a rather high *J*. All the electrocatalytic tests in this work were undertaken without stirring. Moreover, the main reason we chose to report the potential against the SCE is that our current density is high, which would rapidly deplete protons near the electrode, causing a dynamic change in pH. As a result, we believe that providing the type of reference electrode is more accurate. Since OER and HER would occur periodically for the AE process, this work eliminates the need for oxygen-saturated electrolytes. Neither oxygen nor

hydrogen is fed to the electrolyte. We provided the calibration experiments and data in detail (Supplementary Figs. 57–60). In addition to the regular calibration (Supplementary Figs. 57–59), we also provided the relative differences between $Hg/HgCl_2$ electrodes used for electrolysis tests and different reference electrodes (Supplementary Fig. 60). Nickel foam was used directly as the electrode in the two-electrode test. Aqueous electrolyte with 0.2 M $Na^+$ and 60 mM $Co^{2+}$ was used for demonstration.

**Constant potential DFT calculations.** The DFT calculations were carried out in the grid-based projector augmented wave (PAW) formalism as implemented in the GPAW 19.8.1 code[67,68]. The Kohn-Sham equations were solved on a uniform real-space grid with a 0.18 Å grid spacing. The exchange-correlation effects were accounted for by using the BEEF-vdW-functional, which combines the generalized gradient approximation with the Langreth-Lundqvist van der Waals-functional in an optimal way for accurate adsorption energies[69]. The solvent at the electrochemical interface was modeled using a hybrid of implicit/explicit approach[70] combining 36 explicit water molecules around Ni (111), and a SCMVD dielectric continuum model for water[71] was exploited as an implicit solvent to fill the rest. The Ni (111) system was modeled as a $3 \times 3$ supercell in the lateral plane separated by 20 Å of dielectric solvent in between to avoid the interaction between the slab and its period images. The GCE-DFT constant potential calculations in the standard Fermi level ($E_F$) approach were carried out using SJM mode implemented in GPAW[72]. This method achieves constant potential calculations by iteratively adjusting the number of electrons in the simulation cell until the desired $E_F$ is obtained. The simulation cell is kept charge neutral by including a homogeneous counter charge within the implicit solvent regions of the unit cell. The electrode potential is defined as $E_F$ referenced to the electrostatic potential deep in the solvent ($\Phi_w$), where the whole charge on the electrode has been screened, and no electric field is present. The absolute electrode potential ($\Phi_e$) is then determined as

$$\Phi_e = \Phi_w - E_F \tag{1}$$

The absolute electrode potential vs the standard hydrogen electrode $U_{SHE}$ is further defined as below:

$$U_{SHE} = (\Phi_e - \Phi_{SHE})/e \tag{2}$$

where $\Phi_{SHE}$ has been determined experimentally to be ~ 4.44 eV.

The interaction energy ($\Delta E$) is defined as below:

$$\Delta E = E_{tot} - E_{TMs-slab} - (E_{AMCs-slab} - E_{slab} - E_{AMCs}) \tag{3}$$

where $E_{tot}$ is the total energy with both AMCs and TMs (Fe group element ions in this work), $E_{TMs-slab}$ is the total energy with TMs, $E_{AMCs-slab}$ is the total energy with AMCs, $E_{slab}$ is the total energy of pure Ni (111) and $E_{AMCs}$ is the energy of isolated AMCs.

## Constant potential molecular dynamics calculations

The constant potential molecular dynamics (CPMD) can be readily performed for the studies of the dynamics of explicitly solvated electrochemical interfaces as well as for moderately large systems to be investigated for their interfacial properties under constant potential conditions[73]. Hence, MD simulations of electrochemical systems in the GCE can achieve a fully consistent treatment of electrochemical interfaces and reactions. Langevin dynamics (friction = 0.2) was used for the electrochemical interface system to control the temperature (set at 300 K) in the simulations with 1 fs time step and 2 u mass for hydrogen atoms. The trajectories were computed with an eigenstates: $1.0e^{-4}$, density: $1.0e^{-5}$, energy: $1e^{-6}$ setting[74].

The root-mean-square displacement is calculated as below:

$$RMSD = \sqrt{\frac{\sum_{i=1}^{N} d_i^2}{N}} \quad (4)$$

where $d_i$ is the distance between atom i and either a reference structure or the mean position of the N equivalent atoms.

## Data availability
Source data supporting the findings in this study are provided within the paper and its Supplementary Information. Source data are provided with this paper.

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

## Acknowledgements

X.S. acknowledges the funding support from the Free Exploration Project of Frontier Technology for Laoshan Laboratory (No. 16-02). B.T. acknowledges the funding support from the Natural Science Foundation of China (No. 21927811). T.W. acknowledges the funding support from the National Natural Science Foundation of China (No. 52202214). F.L. acknowledges the funding support from the Science and Technology Program of Tibet (No. XZ202201ZY0002G-L.FM). The authors thank BL11B beamline of the Shanghai Synchrotron Radiation Facility (SSRF) for providing the XAFS beamtime. The numerical calculations in this study have been done on Hefei advanced computing center.

## Author contributions

J.Liang conceived the idea, collected and analyzed the data, and wrote the manuscript. Jun Li assisted in electrochemical tests and assisted in analyzing the experimental data. J.Liang. and Z.L. conceived and completed the schematic drawings. T.W. performed the theoretical calculations. H.D., Jiong Li, and T.W. performed the XAFS tests. Y.L. and D.Z. performed the SEM measurements. X.H., Y.W., Y.Y., Y.R., S.S., Q.L., F.L., and G.C. participated in the discussion. X.S. and B.T. supervised this project. All authors contributed and reviewed the manuscript.

## Competing interests

The authors declare no competing interests.
