## [Peer Review File · Nature Communications]

REVIEWER COMMENTS

Reviewer #1 (Remarks to the Author):

Liang et al. performed experimental and theoretical studies to understand the lifespan of electrodes under harsh electrolysis conditions. The authors report an alternating electrolysis approach that realizes promptly and regularly in situ electrode repair/maintenance and concurrent bubble evolution, which successfully extend the electrode lifespan. The AMC-based AE strategy provides a commercial Ni foam electrode with strongly enhanced stability, allowing it to sustain current density of 2000 and -2000 mA cm^{-2} alternately during continuous electrolysis for 93.8h in an acidic solution, whereas the same Ni foam is completely dissolved in just 2.1 h under conventional electrolysis conditions. At the end, the authors performed constant potential & molecular dynamics simulation to understand the role of cations and other influence factors.

1. This is not a pure theoretical work. Instead, this is a work combined experimental and theoretical efforts to understand the science at electrochemical interface. Hence, for the simulation section, the authors should revised some descriptions where emphasizing the comparison between static DFT calculations and molecular dynamics simulation. These comparisons do not really fit this manuscript.

2. The authors only considered the exchange of electron at the electrochemical interface, so-called grand canonical ensemble of electron. It is a better choice to consider the exchange of atom in the framework of grand canonical ensemble monte carlo of atoms, as the dissolution and electro-deposition processes are both grand canonical ensemble of atoms.

3. The authors emphasized multiple times the preliminary results in this work are promising and the improvement is coming. I am quite interested in if the optimization of experimental conditions is able to realize the repair/maintenance of defective electrode? The major concern is the kinetics and time span limitation at AMC-based AE strategy.

4. In the previous experiments, it was found the stability of electrode for electrochemical reaction with large current density, such as OER, there are huge influence factor from mechanical stability, instead of intrinsic chemical stability. In other words, the materials are crashed by the bubbling. If the authors can perform experiments or simulation to distinguish the two aspects? Also which one is more improved in AMC-based AE strategy.

5. In page 26, line 564, there is a typo of "SEH".

Reviewer #2 (Remarks to the Author):

The manuscript by Liang et al. reporting on the alkali metal cations-guided improvement of the electrocatalytic stability of Nickel foam electrode (NiFE) for both the HER and OER at low pH presents some interesting fundamental findings (e.g. in situ formation of protective coatings) but unfortunately is problematic in most key aspects. A range of experimental data are missing, and several key conclusions are highly questionable. The reviewer cannot recommend the publication of this work in a premier journal like Nature Communications, however, would recommend resubmission if the authors can address the comments below.

1| One major criticism the reviewer would like to raise with this manuscript is the reliability of the presented data and discussion thereof.

1.1. First of all, the details provided in the description of the experimental procedures are not at the expected level. It is essentially impossible to understand how the experiments reported in this manuscript were undertaken, leaving minimal chance for other scientists to reproduce any of the results.

Some of the major examples of the numerous flaws of the experimental part are as follows:

--A list of used chemicals/materials is not mentioned; for example, sources from where Nickel foam and Titanium mesh were obtained, and the type of Ag/AgCl reference electrode are not provided.

-- quoting a model of the instrument used for measurements only is, of course, insufficient to describe a physical characterization procedure. Handling and preparation of samples, sample mounting and alignment, all instrumental parameters (e.g. trivial things like scan rate for XRD), key conditions (e.g. level of vacuum, temperature, etc.), data analysis protocols and procedures – all of these must be described.

- It is unclear if the electrocatalytic tests were undertaken with/without stirring, and if the electrolyte solution was/was not saturated with O₂, as it should be. Most critically, how the reference electrode potential was measured? How the "*E*Ag/AgCl (*vs.* NHE)" parameter was determined? Was it precisely measured vs. RHE, as it should be done for a top-quality work, or just a value provided by a manufacturer was assumed to be correct?

Overall, the authors are strongly advised to recognise the importance of the experimental section and essentially completely rewrite it. As an example of almost perfectly written experimental sections, the authors should check papers by Lewis and colleagues, e.g. 10.1039/c7ee01486d.

1.2. The way the data are presented in this paper (no errors attached to the values, discussion of very subtle differences, no information on the repeats of measurements, sometimes an inadequate number of significant figures suggests that only one sample was synthesized, characterized, and tested. The reviewer cannot accept this approach and is confident that no respected journal would accept it either. At least three independent samples of each key material must be synthesized and investigated to ensure the presented data are reliable and reproducible. All key results must be presented as mean \pm one standard deviation, as is normally done.

1.3. The long-term stability experiment and discussion thereof need to be reverified (because there is no reproducibility of data to support the conclusion). In the abstract authors claim that the catalyst can operate without losing its activity for up to 93.8 h (at ± 2 A/cm²). Also, the authors should explain why the measurements were stopped at this hour, is the electrode degrading? This needs to be justified because, by the protective layer concept the electrode/catalyst should never lose its performance especially when all the metal precursors are available for regeneration.

1.4. The main argument when authors compare with the work by Simonov and colleagues (10.1038/s41929-019-0277-8) is that of the toxicity associated with the element Pb. However, there is another work (10.1002/ange.202104123) by the same authors where they explicitly show that the Pb is not required at all for the catalyst to operate in self-healing mode. The authors need to justify not comparing to this work and also suggest expanding their discussion based on this. This eliminates the scheme in Figure 1a.

1.5. One important experiment that needs to be undertaken is the following –

---have the authors tried to coat the NiFe with the protective layer of Co and Na in different electrolytes other than acid? If not please undertake this and then study the stability tests in acid to see if this further enhances the durability of the electrode of interest under HECs.

1.6. Why no survey, C 1s spectra are shown? Without these data, it is impossible to conclude on the quality and reliability of the XPS results.

A significant amount of data collected by the authors is presented in an exceptionally compressed form, avoiding detailed discussions and in-depth analysis of the data. Even the present reviewer found some of the parts of the text challenging to understand. Hence, a reader with a different background in a broad field – the target audience of this journal, will have even fewer chances to understand the work. Overall, essentially every result reported in the manuscript under review requires deeper interpretations, and sometimes further experiments (see below).

2| Electrocatalytic tests.

2.1. First of all, as mentioned above, the reviewer has no confidence in the presented electrochemical data, as the authors do not provide necessary details on the experimental procedures employed, in particular on the reference electrode. In other words, there is no confidence in the reliability of the reported potentials, and therefore in any electrocatalytic data.

A further problem with the potentials is that the values reported herein are against the SCE, and why are they not reported against RHE, when everything is known to authors to convert the potentials? This needs to be undertaken and all the potentials have to be reported against RHE. This is very important keeping in mind that different labs use different reference electrodes.

--Why there are no potential values shown on the y-axis of Figure 3a. Also, nothing is mentioned about the potential either in the discussion or figure captions. This will mislead readers and create confusion.

2.2. The authors are required to undertake EIS measurements and report the effect of different electrolytes on the electrochemical performance, both in terms of stability and activity. Also, would be great if the simulations of the EIS data were undertaken.

2.3. What is the typical volume of electrolyte used for the electrochemical experiments and in this respect how reasonable is the assumption of increased conductivity, which is very much expected?

2.4. Why no Cdl measurements were undertaken by the authors. Also, authors are suggested to undertake similar measurements for the catalyst-free substrate and need to include this discussion during mechanistic studies.

2.5. To further demonstrate the robustness of the approach reported herein, it is required to undertake the stability testing under PEMWE configuration for at least a few hundred hours.

Thus, the paper is problematic in all key scientific aspects – characterization and electrocatalytic tests. Additionally, the description of experimental procedures is inadequate, and the quality of data presentation and discussions is far from perfect. Taken together, all these factors leave the reviewer no choice rather to recommend rejection of any further consideration of this manuscript in its current version.

Point-by-Point Responses to Reviewers' Comments

We sincerely thank the editor and all reviewers for their valuable feedback that we have used to improve the quality of our manuscript (NCOMMS-23-49394). The reviewer comments are shown in *italic*, and our responses, including the added figures, tables, descriptions, and other items, are highlighted in the blue text.

Point-by-point response to the reviewers #1

Reviewer #1 (Remarks to the Author): Liang et al. performed experimental and theoretical studies to understand the lifespan of electrodes under harsh electrolysis conditions. The authors report an alternating electrolysis approach that realizes promptly and regularly in situ electrode repair/maintenance and concurrent bubble evolution, which successfully extend the electrode lifespan. The AMC-based AE strategy provides a commercial Ni foam electrode with strongly enhanced stability, allowing it to sustain current density of 2000 and -2000 mA cm^{-2} alternately during continuous electrolysis for 93.8h in an acidic solution, whereas the same Ni foam is completely dissolved in just 2.1 h under conventional electrolysis conditions. At the end, the authors performed constant potential & molecular dynamics simulation to understand the role of cations and other influence factors.

General Response: We thank the reviewer for the professional comments. According to your suggestions, we have made corrections to our previous draft, and the specifics are provided below.

Comment 1: This is not a pure theoretical work. Instead, this is a work combined experimental and theoretical efforts to understand the science at electrochemical interface. Hence, for the simulation section, the authors should revised some descriptions where emphasizing the comparison between static DFT calculations and molecular dynamics simulation. These comparisons do not really fit this manuscript.

Response 1: Thank you for this comment. As suggested, we have moved simulation calculations comparing different conditions involving electrode potential, molecular dynamics and explicit solvent to the Supplementary Information. After the revision, the theoretical calculation content of our main manuscript has been completely focused on understanding the experimental phenomena.

Comment 2: The authors only considered the exchange of electron at the electrochemical interface, so-called grand canonical ensemble of electron. It is a better choice to consider the exchange of atom in the framework of grand canonical ensemble monte carlo of atoms, as the dissolution and electro-deposition processes are both grand canonical ensemble of atoms.

Response 2: We appreciate this comment. We fully agree that the exchange of atom in the framework of grand canonical ensemble monte carlo (GCMC) of atoms is a better choice to understand the dissolution and electro-deposition processes. The prerequisite of correct GCMC simulation to describe the dissolution and electro-deposition

processes is the existence of a reliable potential model. However, the system of our paper is very complicated involving multiple metals, alkali metal cations and aqueous solvents, and their interatomic potential models do not currently exist in the existing repository (<https://www.ctcms.nist.gov/potentials/>, <https://cmse.postech.ac.kr/html/research/2nnmeam.htm>) and are very challenging to establish.

Considering these difficulties, we employed state-of-the-art grand canonical ensemble of electron (GCE) to investigate why alkali metal cations influence the dissolution and electro-deposition processes of metals at the alternating potentials. Indeed, our calculation results show that the GCE-DFT approach can form the hydrated state of alkali metal cations and thus increase the diffusion rate of interface molecules or atoms. This easier diffusion behavior may make the atoms or molecules dispersed more evenly at electrochemical interface. Furthermore, calculation results show that the Co atom exhibits highest adsorption energy than those of Fe and Ni atom; and there is a moderate interaction energy between Na^+ and Co compared with those of Li^+ and K^+ . Thus, we infer that the high deposition and dissolution rates of Co after adding Na^+ under alternating current are due to the diffusion ability of hydrated AMCs, the moderate interaction between Na^+ and Co and the suitable adsorption strength of Co on the electrode material. This information has been included in the main text.

Finally, we also investigated the three-dimensional and flat Co cluster formation on the Ni (111) after adding Na^+ by using grand canonical ensemble of electron (GCE) (**Supplementary Fig. 45**). Note that the flat Co cluster formation represents the growth of uniform protective coating on the Ni (111) and vice versa. Our calculation results show that the flat Co cluster forms with a formation energy (E_f) of 1.07 eV, which is lower than that three-dimensional Co cluster formation with $E_f = 1.26$ eV. It also suggests that the Co metal prefers to deposit uniformly on the Ni core, consistent with the experiment observation. Therefore, our theoretical simulations using the GCE-DFT method can give a reliable microscopic perspective.

Supplementary Fig. 45 | Three-dimensional (a) and flat (b) Co cluster formation on Ni (111).

Comment 3: The authors emphasized multiple times the preliminary results in this work are promising and the improvement is coming. I am quite interested in if the optimization of experimental conditions is able to realize the repair/maintenance of defective electrode? The major concern is the kinetics and time span limitation at AMC-based AE strategy.

Response 3: We appreciate this comment. According to the characterization results, the coating on electrode surface that is eventually formed is practically amorphous species. Moreover, XANES data suggest that the oxidation state of cobalt in the coatings after different electrolysis times are different. Therefore, at this stage it is hard to precisely replenish the atoms by our strategy. The repair processes in the present work are more likely to form random, amorphous structures.

Although we are currently unable to do so, the maintenance of defective electrode is theoretically possible, as many adjustable parameters (please refer to: *Nat. Synt.* **2**, 172–181 (2023)) of the pulsed AE electrochemistry as well as various solution compositions may enable the repair of the electrode defects at the atomic level of precision.

Comment 4: In the previous experiments, it was found the stability of electrode for electrochemical reaction with large current density, such as OER, there are huge influence factor from mechanical stability, instead of intrinsic chemical stability. In other words, the materials are crashed by the bubbling. If the authors can perform experiments or simulation to distinguish the two aspects? Also which one is more improved in AMC-based AE strategy?

Response 4: Thanks for raising this valuable comment. In fact, AMC-based AE process improves both mechanical stability and chemical stability, and it mainly improve the chemical stability for the following reasons.

Firstly, we have observed in the experiments the damage of violent bubble releases on freshly repaired structures on the electrodes (**Fig. 3b**). Bubble desorption is known to produce stress/tension and volume expansion, which can cause the damage of the electrode (*Adv. Funct. Mater.* **32**, 2107308, (2022)). Therefore, alternating electrolysis tests that operate at ampere-level current densities must lead to high stretch forces and volume expansion. Moreover, due to the same current density operation, the destruction of the electrode structure by bubbles is similar for the different ion combos. Coatings based on Co^{2+} and Na^+ can be present on the electrode surface for a much longer period of time than coatings based on other ion combos. Since Co^{2+} and Na^+ have moderate interaction energy and atomic Co shows suitable adsorption strength on the electrode, the lifespan improvement is more due to the improved chemical stability.

On the other hand, some parts of protective coatings formed may have weak adhesion to the electrode. Such relatively loosely connected structures are more easily carried away by gas bubbles. We have observed that the coating remained in the electrolyte long after it was flushed off by the bubbles, without significant dissolution. Thus, the structural damage to the coating from bubbling is much greater than the coating dissolution from acidity.

Intrinsic chemical stability is more improved by the AMC-based AE strategy in the present work. This does not mean, however, that the coating formed will not dissolve.

Instead, under the ideal conditions, the dissolution and deposition of the coating are in dynamic equilibrium and the coating provides 100% protection to the interior (this includes ideal mechanical stability). In fact, better overall mechanical stability of the electrode is difficult to achieve at this stage, but will be one of our ongoing goals.

Comment 5: In page 26, line 564, there is a typo of "" SEH".

Response 5: Thank you, we have corrected this typo and double-checked the revised manuscript.

Point-by-point response to the reviewers #2

Reviewer #2 (Remarks to the Author): The manuscript by Liang et al. reporting on the alkali metal cations-guided improvement of the electrocatalytic stability of Nickel foam electrode (NiFE) for both the HER and OER at low pH presents some interesting fundamental findings (e.g. in situ formation of protective coatings) but unfortunately is problematic in most key aspects. A range of experimental data are missing, and several key conclusions are highly questionable. The reviewer cannot recommend the publication of this work in a premier journal like Nature Communications, however, would recommend resubmission if the authors can address the comments below.

General Response: We appreciate the reviewer for the insightful comments on our work, which are important for us to improve the quality of our work. More experiment details has been added in our revision work. We have studied all your comments carefully and have made corrections which we sincerely hope meet with approval.

Comment 1: One major criticism the reviewer would like to raise with this manuscript is the reliability of the presented data and discussion thereof.

1.1. First of all, the details provided in the description of the experimental procedures are not at the expected level. It is essentially impossible to understand how the experiments reported in this manuscript were undertaken, leaving minimal chance for other scientists to reproduce any of the results.

Some of the major examples of the numerous flaws of the experimental part are as follows:

--A list of used chemicals/materials is not mentioned; for example, sources from where Nickel foam and Titanium mesh were obtained, and the type of Ag/AgCl reference electrode are not provided.

-- quoting a model of the instrument used for measurements only is, of course, insufficient to describe a physical characterization procedure. Handling and preparation of samples, sample mounting and alignment, all instrumental parameters (e.g. trivial things like scan rate for XRD), key conditions (e.g. level of vacuum, temperature, etc.), data analysis protocols and procedures – all of these must be described.

-- It is unclear if the electrocatalytic tests were undertaken with/without stirring, and if the electrolyte solution was/was not saturated with O₂, as it should be. Most critically, how the reference electrode potential was measured? How the “Ag/AgCl” parameter was determined? Was it precisely measured vs. RHE, as it should be done for a top-quality work, or just a value provided by a manufacturer was assumed to be correct?

Overall, the authors are strongly advised to recognize the importance of the experimental section and essentially completely rewrite it. As an example of almost perfectly written experimental sections, the authors should check papers by Lewis and colleagues, e.g. 10.1039/c7ee01486d.

Response 1: We are grateful to the reviewer for the constructive and valuable comment. As suggested, we have provided as much detail as possible in the revised description of experimental procedures so that the readers know how the experiments were carried out,

and we have rewritten the experimental section. Information on materials used is shown below.

Methods

A list of chemicals/materials

In this work, chemicals were used directly without any purification process. We point out that due to the long experimental period of this work (~two years), some of the reagents purchased were from different manufacturers and batches. For some chemicals, like K_2SO_4 , we used products from different manufacturers, but the results were not significantly different. Sulfuric acid (H_2SO_4 , 95.0%~98.0%) was purchased from a local chemical manufacturer, Chengdu Chron Chemical Co., Ltd. Phosphoric acid (H_3PO_4 , $\geq 85.0\%$) was also purchased from a local chemical manufacturer, Chengdu Jinshan Chemical Test Co., Ltd. Perchloric acid (HClO_4 , 70.0%~72.0%) was purchased from Chengdu Jinshan Chemical Test Co., Ltd. Lithium sulfate monohydrate ($\text{Li}_2\text{SO}_4 \cdot \text{H}_2\text{O}$, AR, 99.0%~100.5%) was purchased from Tianjin Kermel Chemical Reagent Co., Ltd. Sodium sulfate (Na_2SO_4 , AR, 99%) was purchased from Shanghai Aladdin Biochemical Technology Co., Ltd. Potassium sulfate (K_2SO_4) was purchased from Tianjin Fuchen Chemical Reagent Co., Ltd (AR, $\geq 99.0\%$) as well as Chengdu Jinshan Chemical Test Co., Ltd (AR, $\geq 99.0\%$). Iron(III) sulfate ($\text{Fe}_2(\text{SO}_4)_3$) was obtained from Shanghai yuanye Bio-Technology Co., Ltd (AR, 21%~23% for Fe) and Chengdu Chron Chemical Co., Ltd (AR). Manganese sulfate (MnSO_4 , AR, 31%~35% for Mn) was purchased from Shanghai Macklin Biochemical Co., Ltd. Cobalt sulfate heptahydrate ($\text{CoSO}_4 \cdot 7\text{H}_2\text{O}$) was obtained from Chengdu Chron Chemical Co., Ltd (AR, $\geq 99.5\%$) as well as Shanghai Macklin Biochemical Co., Ltd (AR, $\geq 99.0\%$). Nickel sulfate hexahydrate ($\text{NiSO}_4 \cdot 6\text{H}_2\text{O}$) was obtained from Chengdu Chron Chemical Co., LTD (AR, $\geq 98.5\%$) as well as Tianjin Fuchen Chemical Reagent Co., Ltd (AR, $\geq 98.5\%$). Potassium chloride (KCl) was purchased from Shanghai Macklin Biochemical Co., Ltd. We used a water purifier (UPT-1-10T) from Sichuan ULUPURE Ultrapure Technology Co. Ltd. (website: <https://www.ccdup.com/>) for the purification of water, and ultrapure water ($18.25 \text{ M}\Omega \text{ cm}^{-1}$) was used throughout the experiments. Photos of the chemical reagents used in this work are also provided in **Fig. R1** or **Supplementary Fig. 53** for readers to reproduce the results. Nickel foam (**Fig. R2a–c**) with a thickness of 2.0 mm was purchased from Kunshan LONGSHENGBAO Electronic Materials Co., LTD (the company website: <http://www.ksslsb.com/>). Titanium mesh with a purity of $> 99.5\%$. (**Fig. R2d,e**) was purchased from Taobao.com. In addition, all our reference electrodes used in this work are commercial reference electrodes (**Fig. R3**). In this work, saturated calomel electrodes (SCE, CHI 150) as the consumables are stored in a 50-mL centrifuge tube containing saturated potassium chloride solution when not in use for a short period of time (e.g., in two or three days), while we would cover and seal the bottom of the electrode (i.e., the ceramic liquid junction) with a plastic cap when not in use for a longer period of time. Not only do we need to maintain the moisture content of the ceramic core, but we also need to prevent crystallization from clogging the ceramic core.

Material characterizations

Electrolytes will remain on the electrode surface after an electrochemical test. Thus, the electrode after the test was rinsed briefly and slow with ultrapure water and dried in an oven (50 °C) as a sample for further characterizations. All the XRD data were collected using a measurement method of stepping with a start angle of 10 degrees and a termination angle of 80 degrees. The step angle was 0.02 degrees. The tube voltage was 40 kV and the tube current was 30 mA. The data were collected at room temperature (~20 °C). We cut a piece of electrode into smaller sizes (e.g., ~0.2 cm × ~0.2 cm) and pasted them onto conductive tapes on the sample stage for normal SEM characterization. For capturing the cross section views, the electrode was cut along the middle and then fixed vertically to the sample stage with the cross section facing upwards after being gently purged clean with argon gas. A magnetron sputtering apparatus was used to spray platinum on the surface of the sample (operation time: ~3 min) to improve the image quality before proceeding characterizations. The vacuum of the electron microscope sample chamber is generally below 5×10^{-5} bar before the collection of images. The accelerating voltage for SEM testing and energy dispersive spectrometer testing was 5 kV and 10 kV, respectively. A piece of dried electrode sample was cut into smaller pieces, which were immersed in ethanol solution for prolonged sonication. Then, the electrode pieces were taken out to obtain colloid suspension for preparation of TEM characterization samples (dripping nearly 10 μ L solution on Cu grids/carbon). In the first step, a low vacuum level is realized by a mechanical pump, which reduces the pressure from atmospheric pressure to a pressure of 1.33 to 0.133 Pa (0.01 to 0.001 mmHg). Next, a higher vacuum level was realized by an oil diffusion pump, and the pressure was reduced from the low vacuum level to a pressure ranging from 13.33×10^{-4} to 10^{-5} Pa (10^{-5} to 10^{-6} mmHg). In addition, the transmission electron microscope was operated at accelerating voltage of 200 kV. The microscope was operated at accelerating voltage of 120 kV. Samples for the XPS characterization (Thermo ESCALAB 250XI adopting Mg as the excitation source) are kept in nitrogen-filled centrifuge tubes sealed with tape before the testing. Data acquisition at different depths is realized based on operation of different argon sputtering times. The etch depth is converted relative to the specimen and is not the true depth of the sample. The analysis pressure for XPS was 2.42×10^{-9} mBar, and preparation pressure was 3.04×10^{-8} mBar. XAFS spectra at the Co K-edge were recorded at the BL11B beamline of Shanghai Synchrotron Radiation Facility (SSRF). The beam current of the storage ring was 200 mA in a top-up mode. The incident photons were monochromatized by a Si(111) double-crystal monochromator, with an energy resolution $\Delta E/E \sim 1.4 \times 10^{-4}$. The spot size at the sample was $\sim 200 \mu\text{m} \times 250 \mu\text{m}$ (H × V). The energy calibration was performed using a Co reference foil. The reference spectra were measured in the transmission mode, with the ionization chambers filled with N₂.

Electrochemical measurements

Acidic electrolytes including diluted H₂SO₄, H₃PO₄ and HClO₄ solutions. The solution will be replaced off for a newly made one if it is not used up within three days. Moreover, different metal sulfates including Li₂SO₄·H₂O, Na₂SO₄, K₂SO₄, MnSO₄, Fe₂(SO₄)₃, CoSO₄·7H₂O and NiSO₄·H₂O are quantitatively added to the acidic solution for AE. In

short, the electrolyte is H₂SO₄ solution (or H₃PO₄ solution), and different metal sulfates can be added to it as reactants. We also directly provide a photo flowchart (**Fig. R4**) to enable the readers to better understand the experiments. Unless otherwise noted, about 60 mL of acidic electrolyte was used as the electrolyte. We adopt inexpensive and readily available substrates like commercial NiFE and TiME for experiment demonstration in order for more groups to replicate our findings and to conduct more varied research based on this work. The scan rate used for the cyclic voltammetry test was 20 mV s⁻¹ with a sensitivity of 0.1 A V⁻¹. Impedance measurements were carried out in different electrolytes without stirring. The high Frequency was fixed at 100,000 Hz and the low frequency was fixed at 0.01 Hz. The initial E was equal to the open-circuit potential for each test, and the amplitude was fixed at 0.003 V. Electrochemical experiments were measured using a CHI 660E workstation (or CHI 760E) and a three-electrode system. We employed a commercial graphite rod (replaced on a regular basis) as the counter electrodes to exclude unwanted interference with metal leaching from the metal counter electrodes. The reference electrode, saturated calomel electrode (SCE) or Ag/AgCl, is also replaced after a certain period of electrolysis. No *iR* correction is performed because the open circuit potential actually changes all the time and the electrochemical test is kept at rather high *J*. All the electrocatalytic tests in this work were undertaken without stirring. Moreover, the main reason we chose to report the potential against the SCE is because our current density is really high, which would rapidly deplete protons near the electrode, causing a dynamic change in pH. As a result, we believed that providing the type of reference electrode is more accurate. Since OER and HER would occur periodically for the AE process, this work eliminates the need for oxygen-saturated electrolytes. Neither oxygen nor hydrogen is fed to the electrolyte.

We understand the reviewer's views regarding the potentials based on SCE or RHE scale. In the first submission, we did not convert the electrode potential to the RHE scale because of several vital reasons. First of all, the main reason we chose to report the potential against the SCE in the first submission is because our current density is really high (e.g., up to 2 A cm⁻²), which would rapidly deplete protons/hydroxide ions from water near the electrode (please refer to: *Joule* **8**, 728–745 (2024)), causing a dynamic change in pH. As a result, we believed that providing the reader with the type of reference electrode might be more useful and accurate in our first submission. Moreover, the pH of the electrolyte changes after a long period of electrolysis (our measurements show that the pH usually rises by about 0.2~0.4 after the electrolysis for the best Co²⁺-Na⁺ combo), and this can make a pH-based RHE scale ($E_{(vs.RHE)} = E_{(vs.ref.)} + E^{\theta}_{(ref.)} + 0.059 \cdot pH$) more inaccurate instead. In addition, the focus of this work is on extending the lifespan of the electrode, so the potential information is directly based on the reference electrode (e.g., XX.X V vs SCE or vs Ag/AgCl). For above reasons, we did not calibrate the reference electrode in the first submission. While reporting the potential based on the SCE scale is reasonable, we do understand your concern and appreciate the comments. As suggested, we also provided the potentials against the RHE scale in the revised Supplementary Information (**Supplementary Figs. 14–17**) to address your concerns. Also, we have provided related calibration experiments and data in detail (**Figs. R5–8**). In addition to the regular calibration (**Figs. R5–7**, please refer to

ACS Energy Lett. **5**, 1083–1087 (2020)), we also provided the relative differences between SCE and different reference electrodes (**Fig. R8a**). The open-circuit potential tests of different SCE electrodes shows that the voltage differences between the reference electrode after a period of use and the new reference electrode are only a few tenths of a millivolt (**Fig. R8b**).

Indeed, we agree that high purity O₂ bubbled through the electrolyte can saturate it and fix the reversible oxygen potential during the oxygen evolution reaction (OER) process (*Chem. Mater.* **29**, 120–140 (2017), *ACS Catal.* **8**, 9359–9363 (2018)), but our work is genuinely special because the alternating electrolysis process, where OER and hydrogen evolution reaction (HER) would occur periodically on the electrode surfaces, renders such saturation operation less meaningful. Moreover, our AE processes shows very high current densities. In fact, neither oxygen nor hydrogen is fed to the electrolyte for two-electrode electrolysis production. Therefore, our work eliminates the need for oxygen-saturated electrolytes. Thank you again for your kind reminder, we have added this information to the revised manuscript.

In short, we have carefully rewritten the experimental section in accordance with your kind suggestions and in light of the paper by Lewis and colleagues (e.g. *Energy Environ. Sci.* DOI: 10.1039/c7ee01486d) for describing the experimental details so that readers can repeat our experiments. The discussion in the Response 1 for Comment 1 has been added to the revised manuscript and Supplementary Information to make our findings easier for readers to repeat. Moreover, we added **Supplementary Note 1** and **Supplementary Note 2** to provide more details/messages about this work. In fact, the experimental details we have given are actually far more than a lot of recent studies now.

Fig. R1 Photos of main chemical reagents used for the experiments in this work.

Fig. R2 Photos of the metal substrates used in this work. (a,b) Photos of the nickel foam and (c) the corresponding enlarged image providing more details. (c) Photo of the titanium mesh and the corresponding (d) enlarged image. Enlarged images of nickel foam and copper foam are not included with scale bars because they were taken via common magnifying lens.

Fig. R3 Photos of reference electrodes. (a) Saturated calomel electrode purchased from CH Instruments Ins (the website: <http://www.chinstr.com/sy>). (b) Ag/AgCl electrode purchased from CH Instruments Ins. (c) Sulfate reference electrode purchased from CH Instruments Ins. (d) Sulfate reference electrode purchased from SHANGHAI CHUXI Industrial Co., Ltd (the website: <http://www.chuxi17.com/>).

Fig. R4 (a) Photos of the acidic H_2SO_4 solution for preparing the electrolyte. (b) Photos of the acidic solution with Na^+ and Co^{2+} . (c) Undivided electrolyzer for screening ion combinations.

Fig. R5 Voltammery curves for the calibration of $\text{Hg}/\text{Hg}_2\text{Cl}_2$ electrodes in (a) H_2SO_4 solution, (b) H_2SO_4 solution with 1 M Li^+ , (c) H_2SO_4 solution with 1 M Na^+ , and (d) H_2SO_4 solution with 1 M K^+ (solution $\text{pH} > 1$). Experimental procedures for correcting the reference electrodes can be find in previous literature (e.g., *ACS Energy Lett.* **5**, 1083–1087 (2020), *ACS Catal.* **13**, 1893–1898 (2023), and *Nat. Mater.* **10**, 780–786 (2011)). The fluctuations in the curves are due to the intense H_2 bubble flow (**Fig. R6b**). The limited geometry surface area and the limited active sites of the Pt plate should be

the reasons for the small currents of the polarization curves. Besides, we used the RHE calibration data of Hg/Hg₂Cl₂ electrode obtained in solutions without the addition of iron group element ions to calibrate the data for all solutions with the addition of iron group element ions because (i) the difference in pH value is not significant and (ii) the metal ions may be deposited in the potential range of the CV tests.

Fig. R6 (a) Photo of the Pt plate electrodes, a gas tube, and a reference electrode. (b) Rapid release of H₂ bubbles from the gas tube during correction tests. (c) Hydrogen generator used in this work.

Fig. R7 Voltammetry curve for the calibration of Hg/Hg₂Cl₂ electrodes in 0.1 M HClO₄ solution. To provide more information of the Hg/Hg₂Cl₂ electrode, we also provide the calibration data in HClO₄ solution of known a concentration in that HClO₄ ionizes 100% in solution.

Fig. R8 (a) Polarization curves of a commercial Pt plate electrode recorded with different reference electrodes in the same acidic electrolyte. The potentials are voltages recorded directly by the electrochemical workstation. (b) Time-dependent open-circuit potential values. For the measurements in **Fig. R8b**, connecting a new SCE electrode to the reference electrode clamp and counter electrode clamp, and connecting the SCE electrodes after hundreds of hours of use to the working electrode clamp. SCE-1 was used for about 200 h and SCE-2 was used for about 150 h. The tests further indicate that the used SCE electrodes were not damaged.

Comment 2: 1.2. The way the data are presented in this paper (no errors attached to the values, discussion of very subtle differences, no information on the repeats of measurements, sometimes an inadequate number of significant figures suggests that only one sample was synthesized, characterized, and tested. The reviewer cannot accept this approach and is confident that no respected journal would accept it either. At least three independent samples of each key material must be synthesized and investigated to ensure the presented data are reliable and reproducible. All key results must be presented as mean \pm one standard deviation, as is normally done.

Response 2: Thank you for the helpful advice. Firstly, a large amount of our data are electrochemical stability tests (like data in **Fig. 2** and **Fig. 3**), which do not usually require error bars (for instance, please refer to *Nature* **617**, 519–523 (2023) (Figure 2a), *Nature* **618**, 959–966 (2023) (Figure 3g), *Nat. Mater.* **22**, 1022–1029 (2023) (Figure 5c), *Nat. Energy* **8**, 264–272 (2023) (Figure 5d), *Nat. Commun.* **15**, 1767 (2024) (Figure 5d), and *Nat. Mater.* **22**, 100–108 (2023) (Figure 3f)). We have conducted almost two years of testing work to complete the data in the manuscript, and it would take a great deal more time to obtain the error bars for such tests. Having said that, we do understand your concerns and appreciate your comments.

Our key results in this work are the longest electrode lifespan of 93.8 h based on the best Na^+ - Co^{2+} combo. As suggested, we have provided the key results (93.8 h lifespan) as mean \pm one standard deviation (**Fig. R9**). In addition, we also obtained the longer electrode stability when the current densities of the AE process were changed to -1 & 1 A cm^{-2} (99.2 h) (please see **Fig. R10**), again illustrating the feasibility of our method

and the effective combination of Na^+ and Co^{2+} towards lifetime enhancement.

Fig. R9 Data of five independent stability measurement results to ensure the reliability and reproducibility of our key findings. (a) AE tests in the acidic solution based on the optimal Na^+ - Co^{2+} combo. (b) Average electrolysis time length. The average electrolysis lifespan is 93.4 ± 8.2 h. The error bar represents the results of different independent tests.

Fig. R10 (a) AE data based on the Co^{2+} - Na^+ combo. Since the test current densities were reduced to -1 & 1 A cm^{-2} , we reduced the corresponding Co^{2+} content to 0.2 M . Na^+ levels remain in excess (1 M).

Comment 3: 1.3. The long-term stability experiment and discussion thereof need to be recertified (because there is no reproducibility of data to support the conclusion). In the abstract authors claim that the catalyst can operate without losing its activity for up to 93.8 h (at $\pm 2 \text{ A/cm}^2$). Also, the authors should explain why the measurements were stopped at this hour; is the electrode degrading? This needs to be justified because, by

the protective layer concept the electrode/catalyst should never lose its performance especially when all the metal precursors are available for regeneration.

Response 3: Firstly, thank you for the comment on data reproducibility to support the conclusion again. We have provided different independent test results to support the key conclusion (please see Response 2).

Secondly, we have never claimed in the Abstract or even in full manuscript that the catalyst can operate without losing its activity for up to 93.8 h (at $\pm 2 \text{ A/cm}^2$). Please note that we are not claiming no loss of activity. Our exact sentence in the Abstract is: noticeably, our AMC-based AE strategy provides a commercial Ni foam electrode with strongly enhanced stability, allowing it to sustain J of 2000 and -2000 mA cm^{-2} alternately during continuous electrolysis for 93.8 h, whereas a Ni foam is completely dissolved in just 2.1 h under conventional electrolysis conditions.

Furthermore, the primary cause of the measurement's stopping after 93.8 h is the fact that the inside nickel foam is still corroding bit by bit during electrolysis, despite the good protection by our strategies. In other words, we are still not able to protect the nickel foam 100% from exposure to the acids. The fact that nickel foam is running at extremely high current densities in an acidic solution makes it impossible to avoid coming into contact with the acid. We set the maximum voltage limits (10 & -10 V) so that the electrochemical station program will automatically stop once the voltage required to achieve the current densities of 2 A cm^{-2} & -2 A cm^{-2} are too high. As shown in **Fig. R11**, the applied potential is gradually increasing during the test, and finally a potential of 10 V can no longer support the electrode to attain a current density of 2 A cm^{-2} . At this point, electrolysis is terminated.

We report the important discovery of the co-action of Fe group elemental ions and alkali metal cations. Also, it is vital to emphasize that it was not easy to realize the 93.8 h lifespan of Ni foam under high current densities of 2 A cm^{-2} & -2 A cm^{-2} in acidic solution. We have provided more detailed explanations for why the measurement was stopped at 93.8 h, which can be found below or in **Supplementary Notes 2**.

Supplementary Notes 2 | Detailed reasons for the final degradation of performance as well as why permanent alternating electrolysis cannot be achieved at this stage.

- 1) Reason 1: We set maximum voltage limits (10 V & -10 V) so that the electrochemical station program will automatically stop once the voltage required to achieve the current densities of 2 A cm^{-2} & -2 A cm^{-2} are too high. As shown in **Fig. R11**, the applied electrode potentials are gradually increasing during the test, and finally a potential of 10 V can no longer support the electrode to attain the current density of 2 A cm^{-2} . At this point, electrolysis is terminated.
- 2) Reason 2: The fact that nickel foam is running at extremely high current densities in an acidic solution makes it impossible to avoid coming into contact with the acids. Even with the external/outer protective layer, the internal nickel foam is still more or less etched by the acidic solution. At the end of our tests, sometimes the internal nickel foam had dissolved but the external protective layer was still there. This isolates the wires inside the electrode holder from the electrode. This breaks the circuit and thus stops the electrolysis process.

- 3) Reason 3: Generally, the longer the electrolysis time, the thicker the external/outer protective layer becomes, and the excessive thickness of the protective layer is one of the reasons for the decay of activity. This is because the proper coating thickness is still difficult to maintain at this stage.
- 4) Reason 4: Due to the technological limitations, this deposition-dissolution process is not fully reversible, which is similar to the charging and discharging process of a battery.

Fig. R11 AE data of the NiFe. Electrolysis is forced to stop at 93.8h (see the red frame) when the required voltage was too high.

Comment 4: 1.4. The main argument when authors compare with the work by Simonov and colleagues (10.1038/s41929-019-0277-8) is that of the toxicity associated with the element Pb. However, there is another work (10.1002/ange.202104123) by the same authors where they explicitly show that the Pb is not required at all for the catalyst to operate in self-healing mode. The authors need to justify not comparing to this work and also suggest expanding their discussion based on this. This eliminates the scheme in Figure 1a.

Response 4: We greatly appreciate the insightful feedback from the reviewer. We fully understand your concern. We chose this work in **Fig. 1a** for comparison not only because it reports one of the most effective and representative strategies, but also because metal ions was added to the electrolyte in that work (same in our work).

First of all, we appreciate your comments and have carefully studied the literature you mentioned. The work you mentioned (10.1002/ange.202104123, title: Stable acidic water oxidation with a cobalt–iron–lead oxide catalyst operating via a cobalt-selective self-healing mechanism) is strictly different from the work we compared in **Fig. 1a**. The work you mentioned reported a Pb-containing catalyst, [Co–Fe–Pb]O_x, operated in acidic electrolytes in the absence of dissolved Pb²⁺, while the work by Simonov and colleagues demonstrated the use of Pb²⁺ in solution to improve electrode stability. Either way, Pb was used in the two studies. Instead of indicating that Pb is not required for self-healing processes, the work you mentioned (10.1002/ange.202104123) again proves the importance of Pb for improving electrode stability in acidic solution.

In fact, many previous studies (for instance, please refer to *Chem. Sci.* **8**, 4779–4794 (2017) and *Proc. Natl Acad. Sci.* **117**, 16187–16192 (2020)) have demonstrated that Pb is an effective element to enhance the electrode stability in acidic solution, whether it is introduced into the catalyst at the first place (the work you mentioned) or added to

the electrolyte later (the work in **Fig. 1a**), which involves the use of Pb, a known toxic element.

In addition, one important fact is that the current densities achieved by their work are much smaller than those achieved by our work. Other vital point is that more parameter adjustments are possible with the AE method we employed than with the conventional electrolysis approach. Indeed, there are several strategies for achieving catalyst self-healing (e.g., *Angew. Chem. Int. Ed.* **60**, 15821 (2021)). However, the AE-based methods in our study not only establish a new reliable strategy to repair electrode but also paves the way for the development of AE-based systems for the production of value-added chemicals, AMC-based electrocatalytic systems, and AMC-based electrodeposition technique, and beyond. Therefore, we do have enough good reasons to make comparisons in **Fig. 1**. Finally, we thank you for the professional comment.

As suggested, we have expanded the discussion based on this (**Supplementary Note 1**).

Comment 5: 1.5. One important experiment that needs to be undertaken is the following –have the authors tried to coat the NiFE with the protective layer of Co and Na in different electrolytes other than acid? If not please undertake this and then study the stability tests in acid to see if this further enhances the durability of the electrode of interest under HECs.

Response 5: We thank the reviewer for the very helpful comments. First of all, the electrodes do not corrode in alkaline environments. Current alkaline water electrolysis technologies are mature. Metal electrodes are stable enough in alkaline environments, so there is no need for alternating electrolysis; conventional electrolysis is just fine. In general, metal electrodes only corrode in acidic environments, which is why we decided to develop this method. Therefore, the main focus of this work is to improve the lifespan of electrodes in acidic solutions.

Having said that, we understand that you're wondering if the protective layer of Co and Na is still effective at other pH levels. As suggested, we tried to coat the NiFE with the protective layer of Co and Na in solution with a higher pH (~4.4). The results shows that the lifetime of the NiFE can be easily boosted to at least 180 h (**Fig. R12**). More importantly, the applied potentials were still at a low level. This indicates that we have not yet hit the maximum rise in life expectancy. We artificially stopped this test.

In addition, studies were carried out to more thoroughly demonstrate the interaction between cobalt and sodium that may lead to better stability in acids. We have coated Ti mesh with Co_3O_4 ($\text{Co}_3\text{O}_4/\text{TM}$) and Ti mesh with Na-doped Co_3O_4 ($\text{Na-Co}_3\text{O}_4/\text{TM}$) according to a previous work (*Appl. Catal. B* **317**, 121769 (2022)) and studied the stability performance in acidic media. Interestingly, even the direct Na doping enhances the stability of Co_3O_4 , with results showing that the $\text{Na-Co}_3\text{O}_4/\text{TM}$ achieves 15.3 times the electrolysis time of the $\text{Co}_3\text{O}_4/\text{TM}$ (4.75 h versus 0.31 h, **Fig. R13**). This result further affirms the significance of our work, i.e., alkali metals and iron group metals may not only synergize during the alternating electrolysis tests in our present work, but better electrocatalytic performance in acidic solution may also be achieved by preparing them directly as the electrocatalysts.

Fig. R12 AE data based on the Co^{2+} - Na^+ combo in H_2SO_4 solution with a relatively high pH value of ~ 4.4 . The current densities were reduced to around -1 & 1 A cm^{-2} , and the corresponding Co^{2+} content was reduced to 0.2 M . Na^+ levels remain in excess (1 M). The reasons for not converting potentials to RHE dependence here have already been given in the Response 1.

Fig. R13 (a) Polarization curves of different electrodes in H_2SO_4 solution ($\text{pH} = 1$). (b) Stability comparison of $\text{Co}_3\text{O}_4/\text{TM}$ and $\text{Na-Co}_3\text{O}_4/\text{TM}$.

Comment 6: 1.6. Why no survey, C 1s spectra are shown? Without these data, it is impossible to conclude on the quality and reliability of the XPS results.

Response 6: We understand your concerns and agree that our manuscript would benefit from a more detailed XPS results. As suggested, we have provided all survey spectra, the C 1s spectra and the original data (**Fig. R14-R16**).

Fig. R14 (a) XPS survey spectra of the post-reaction electrodes after 20-h AE tests. (b-c) Depth-profiling XPS spectra in the C 1s region, with both original data curves and related fitting data curves.

Fig. R15 (a) XPS survey spectra of the post-reaction electrodes after 30-h AE tests. (b-c) Depth-profiling XPS spectra in the C 1s region, with both original data curves and related fitting data curves.

Fig. R16 (a) XPS survey spectra of the post-reaction electrodes after 50-h AE tests. (b-c) Depth-profiling XPS spectra in the C 1s region, with both original data curves and related fitting data curves.

Comment 7: A significant amount of data collected by the authors is presented in an exceptionally compressed form, avoiding detailed discussions and in-depth analysis of the data. Even the present reviewer found some of the parts of the text challenging to understand. Hence, a reader with a different background in a broad field – the target audience of this journal, will have even fewer chances to understand the work. Overall, essentially every result reported in the manuscript under review requires deeper interpretations, and sometimes further experiments (see below).

2 | Electrocatalytic tests.

2.1. First of all, as mentioned above, the reviewer has no confidence in the presented electrochemical data, as the authors do not provide necessary details on the experimental procedures employed, in particular on the reference electrode. In other words, there is no confidence in the reliability of the reported potentials, and therefore in any electrocatalytic data.

A further problem with the potentials is that the values reported herein are against the SCE, and why are they not reported against RHE, when everything is known to authors to convert the potentials? This needs to be undertaken and all the potentials have to be reported against RHE. This is very important keeping in mind that different labs use different reference electrodes.

--Why there are no potential values shown on the y-axis of Figure 3a. Also, nothing is mentioned about the potential either in the discussion or figure captions. This will mislead readers and create confusion.

Response 7: We sincerely appreciate the reviewer for the insightful comments on our work. We incorporate a lot of data in **Fig. 3a** in the first submission for the purpose of better demonstrating trends. As suggested, we also provided the data of **Fig. 3a** in uncompressed form in the revised Supplementary Information to address your concern (please see **Figs. R17-20**). Besides, in order for readers to better understand the work, we have added the more experimental details in the revised work.

Moreover, the main reason we chose to report the potential against the SCE in the first submission is mainly because our current densities are really high, which would rapidly deplete protons/hydroxide ions from water near the electrode (please refer to: *Joule* **8**, 728–745 (2024)), causing a dynamic change in pH. More reasons can be found in **Response 1**. As a result, we believed that providing the reader with the type of reference electrode and the pH of the solution might be more useful and accurate in our first submission. We do understand your concern. As suggested, we also provided the potentials against the RHE scale in the revised Supplementary Information to address your concerns.

In addition, the electrode potential values of **Fig. 3a** were actually provided in our first submission of the manuscript (please see the red circle in **Fig.R21**). It's possible that you missed it because we didn't make it obvious. We plotted it in the **Fig. 3a**'s lower right corner in the hope that it would be a more visually appealing image. We now realize the mistake. As suggested, **Fig.3a** with potential values shown on the y-axis are also provided now in the revised manuscript (please see **Fig.R22**) and revised Supplementary Information (please find the data with the potential values again the RHE in **Fig.R23–26**) to address your concerns. Again, we appreciate your helpful suggestions for the presentation of data and images.

Fig. R17 Data of the second row of Fig. 3a in uncompressed form, showing alternating electrolysis results based on Fe group element ions.

Fig. R18 Data of the second row of Fig. 3a in uncompressed form, showing alternating electrolysis results based on Li^+ alone and results based on Fe group element ions and Li^+ .

Fig. R19 Data of the third row of Fig. 3a in uncompressed form, showing alternating electrolysis results based on Na^+ alone and results based on Fe group element ions and Na^+ .

Fig. R20 Data of the fourth row of Fig. 3a in uncompressed form, showing alternating electrolysis results based on K^+ alone and results based on Fe group element ions and K^+ .

Fig. R21 Original Fig. 3a in the first submission. The potential information was already in this Figure.

Fig. R22 Revised Fig. 3a with the clear potential values. The potentials reported in this figure are relative to the SCE.

Fig. R23 Data of the second row of Fig. 3a in uncompressed form, showing alternating electrolysis results based on Fe group element ions. The potentials reported in this figure are relative to the reversible hydrogen electrode (RHE).

Fig. R24 Data of the third row of Fig. 3a in uncompressed form, showing alternating electrolysis results based on Li⁺ alone and results based on Fe group element ions and Li⁺. The potentials reported in this figure are relative to the reversible hydrogen electrode (RHE).

Fig. R25 Data of the third row of Fig. 3a in uncompressed form, showing alternating electrolysis results based on Na^+ alone and results based on Fe group element ions and Na^+ . The potentials reported in this figure are relative to the reversible hydrogen electrode (RHE).

Fig. R26 Data of the third row of Fig. 3a in uncompressed form, showing alternating electrolysis results based on K^+ alone and results based on Fe group element ions and K^+ . The potentials reported in this figure are relative to the reversible hydrogen electrode (RHE).

Comment 8: The authors are required to undertake EIS measurements and report the effect of different electrolytes on the electrochemical performance, both in terms of stability and activity. Also, it would be great if the simulations of the EIS data were undertaken.

Response 8: We understand this is a good suggestion, and we agree that electrochemical impedance spectroscopy (EIS) data would provide useful insights. Due to continuous dissolution issues of Ni foam and coatings in acidic solutions, the EIS data comparison was recorded in 0.5 M Na_2SO_4 solution. In fact, under acidic conditions, a lot of the ion combinations have difficulty to build up a protective layer on the nickel surface. Since no protective layer is formed, the resulting anodes remain virtually unchanged after 1 min of electrolysis tests (**Fig. R27a**). For example, in the acidic electrolyte containing Li^+ and Ni^{2+} (NF-2, see **Fig. R27a**) as well as K^+ (NF-3, see **Fig. R27a**), the electrodes did not show significant changes based on EIS plots. Simply adding Na^+ (NF-4) or Na^+ plus Ni^{2+} (NF-5) will not have a significant change. Please note that even though some combinations can form a protective layer on the surface of the nickel foam during the repair stage, eventually such a protective layer will be shattered (please see the second row in **Fig. 3b**), so that the surface of the nickel foam is not protected in any way and

can only be continuously etched by the acid. Since ion combinations, like $\text{Li}^+\text{-Ni}^{2+}$ and $\text{Na}^+\text{-Ni}^{2+}$, can hardly generate the protective layer on the inner Ni core, the best combination of Na^+ and Co^{2+} was adopted to demonstrate the changes brought about by electrolysis on the Ni surface to address your concerns. Different electrolysis programs were performed to illustrate the effect of effect of $\text{Na}^+\text{-Co}^{2+}$ -based electrolytes on the electrochemical performance. This, we believe, is of high importance to better understand the best performance based on the $\text{Na}^+\text{-Co}^{2+}$ combo. First of all, compared to Nyquist plots of NF-1 before electrolysis (**Fig. R27a**), the EIS semicircle becomes smaller after a 10-min OER process (**Fig. R27b**) in the acidic solution with Co^{2+} and Na^+ , thus indicating the anodic bias can oxidize NF, but this semicircle change is not as significant as the change after the HER process. Well-separated semicircles can be observed for the NF after a 10-min HER process under different testing parameters (**Fig. R27c**). This suggests that there is a marked difference between the changes made to the electrodes by HER and OER electrolysis alone. Also, running at the anodic potentials does not introduce cobalt-containing catalytic materials. In addition, the data in **Fig. R27d-h** clearly demonstrate that longer AE testing times would lead to smaller EIS semicircles, thus confirming faster charge transfer processes. Since the data we recorded are of high quality, the change in electrochemical behavior is evident in the enlarged detail. For instance, the resistance of charge transfer achieved by the HER process alone is significantly smaller than that achieved by the AE process at the same time. The charge-transfer resistance (R_{ct}) achieved by the AE process alone is significantly smaller than that achieved by the OER process at the same time. In addition, extending the test time for AE leads to more species on the catalyst surface (**Fig. R27h**), which matches the characterization results of HRTEM images. Passivation film may exist on the surface of the Ni electrode after 3 h of electrolysis (**Fig. R27g**), the migration process of ions is inhibited in the presence of the film resistance, so that the impedance spectrum in the low-frequency part also exhibits a line with a 45° inclination. This phenomenon has some differences compared to the impedance data recorded on the sample after 22.8 h of electrolysis. After a sufficiently long period of tests, when the AC signal passes through the electrode, diffusion control will override the electrochemical control in the low-frequency portion, and the Warburg impedance will appear (**Fig. R27h**). The fitting data for sample recorded after 10-min HER testing, after 10-min AE testing, samples after 1.25-h AE testing, sample after 3-h AE testing, and the sample after 22.8-h AE testing can be found in **Fig. R28**. The insets provide the equivalent circuits and detailed impedance fitting data.

Fig. R27 (a) Independent EIS data of five pieces of Ni foam (NF) electrodes before and after testing in different acidic H₂SO₄ solution with different ions, including NF-1, NF-2, NF-3, NF-4, and NF-5. (b) Comparison of the EIS data before electrolysis and after conventional OER/HER electrolysis. Tests for the HER employed different low-frequency parameters, so there are two impedance results (the impedance arc is basically the same). (c) Magnification of details. (d) Comparison of the EIS data before electrolysis and after AE for 10 min and 12.5 h. (e) Magnification of details. (f) Comparison of the EIS data before electrolysis and after AE for 10 min, 12.5 h, 3 h, and 22.8 h. (g) Magnification of details. (h) Magnification of details.

Fig. R28 (a) EIS fitting results for sample after 10-min HER testing. (b) EIS fitting results for sample after 10 min of AE testing. (c) EIS fitting results for sample after 1.25-h AE testing. (d) EIS fitting results for sample after 3-h AE testing. (e) EIS fitting results for sample after 22.8-h AE testing. The equivalent circuit used for modeling the measured electrochemical response are given as inset images. CPE stands for constant phase angle element. Rct, R1, R2, and R3 are related with the kinetics of the interfacial charge transfer reaction. R_s represents the solution resistance (*J. Am. Chem. Soc.* **142**, 12087–12095 (2020), *Nat. Commun.* **14**, 1873 (2023), *Electrochim. Acta* **418**, 140350 (2022)).

Comment 9: What is the typical volume of electrolyte used for the electrochemical experiments and in this respect how reasonable is the assumption of increased conductivity, which is very much expected?

Response 9: We appreciate this comment. First of all, the typical volume of electrolyte used for the experiments is 50 mL. In addition, we understand your concerns. In many cases, better conductivity can result in improved electrochemical performance. In this study, however, the conductivity is not a decisive factor in determining the efficacy of the electrode lifespan enhancement. To demonstrate this, we measured and compared the solution resistance values of three different sets of electrolytes (**Fig. R29**), namely the electrolyte containing 1 M Na^+ and 0.3 M Co^{2+} , the electrolyte containing 1 M K^+ and 0.3 M Co^{2+} , and the electrolyte containing 1 M Na^+ and 0.3 M Ni^{2+} . Based on these ion combinations, the realized electrode lifespan varies considerably. Noticeably, the solution resistance values corresponding to the three ion combinations do not differ significantly from each other (**Fig. R29c**), and the solution resistance of the best ion combination (i.e., 5.85 Ω for the Na^+ - Co^{2+} combo) is even slightly larger than that of the other two combinations (5.10 Ω for the K^+ - Co^{2+} combo and 4.42 Ω for the Na^+ - Ni^{2+} combo). Therefore, the data confirm that the increased conductivity does not lead to a significant increase in the lifespan of the electrode.

Fig. R29 (a) Comparison of Nyquist plots in different electrolytes. (b) Magnification of details. (c) Comparison of the solution resistance values. All the three independent tests used the same the volume of electrolyte, the same working electrode, the same reference electrode, and the same counter electrode. The only variable was the ionic species of the electrolyte.

Comment 10: Why no Cdl measurements were undertaken by the authors. Also, authors are suggested to undertake similar measurements for the catalyst-free substrate and need to include this discussion during mechanistic studies.

Response 10: We thank the reviewer for this helpful comment. As suggested, we have performed C_{dl} measurements to study the changes of the electrode. We used the best combination of ions, i.e., Co²⁺ and Na⁺, to perform the AE tests. The C_{dl} value of the resulting electrode does rise after a short period of electrolysis (**Fig. R30**), indicating an increase in the electrochemical active surface area (ECSA). This actually matches the results of the SEM characterization and electrochemical data, where the protective coatings with catalytic activity appear on the substrate surface after electrolysis based the Co²⁺-Na⁺ combo.

Therefore, the enlarged ECSA should lead to better electrochemical activity.

Fig. R30 (a) C_{dl} value of Ni foam before electrolysis. The inset shows the corresponding cyclic voltammograms at scan rates of 10, 30, 50, 70, 90, and 110 mV s⁻¹. (b) C_{dl} value of Ni foam after a 10-min AE process and the related cyclic voltammograms. (c) C_{dl} value of Ni foam after a 30-min AE process and the related cyclic voltammograms. All electrochemical tests were recorded in 0.5 M Na₂SO₄ aqueous electrolyte, since acidic solution would consistently corrode the electrodes to affect the comparison of results.

Comment 11: To further demonstrate the robustness of the approach reported herein, it is required to undertake the stability testing under PEMWE configuration for at least a few hundred hours.

Response 11: Thank you. As suggested, we have performed the stability testing under a PEMWE configuration (**Fig. R31**). We achieved over 250 hours of operation without the use of any precious metals.

Fig. R31 PEM-based cell tests under room temperatures. (a) Photo of the assembled PEM water electrolyzer. (b) Comparison of electrolysis time length under two different conditions. The applied cell voltage is 2.8 V. The electrochemical tests were recorded in aqueous electrolyte with 0.2 M Na⁺ (an excess content) and 60 mM Co²⁺.

Thus, the paper is problematic in all key scientific aspects – characterization and electrocatalytic tests. Additionally, the description of experimental procedures is inadequate, and the quality of data presentation and discussions is far from perfect. Taken together, all these factors leave the reviewer no choice rather to recommend rejection of any further consideration of this manuscript in its current version.

Response: Thank you for your valuable comments. We have studied all your comments carefully and have made corrections, including characterization, electrocatalytic tests, experimental procedure descriptions, and the quality of data presentation and discussions. We believed the revised version is ready for acceptance.

REVIEWERS' COMMENTS

Reviewer #1 (Remarks to the Author):

The authors frankly respond to my concerns and explain in more details for the experimental phenomena. I think this work is publishable.

Reviewer #2 (Remarks to the Author):

The authors have done a wonderful job in addressing all the important concerns that were raised previously, especially the detailed experimental section and the reproducibility of the results. The reviewer thank the authors for taking the experimental section seriously and coming out with an elaborate description, this was very much essential in today's science, where reproducibility is a big concern.

The manuscript can now be accepted in its current form.